# A 1-km global dataset of historical (1979-2013) and future (2020-2100) Köppen-Geiger climate classification and bioclimatic variables

Diyang Cui[1], Shunlin Liang[1], Dongdong Wang[1], Zheng Liu[1]

[1]Department of Geographical Sciences, University of Maryland, College Park, 20740, USA

*Correspondence to*: Shunlin Liang(sliang@umd.edu)

**Abstract.**

The Köppen-Geiger classification scheme provides an effective and ecologically meaningful way to characterize climatic conditions and has been widely applied in climate change studies. Significant changes in the Köppen climates have been observed and projected in the recent two centuries. Current accuracy, temporal coverage, spatial and temporal resolution of

historical and future climate classification maps cannot sufficiently fulfil the current needs of climate change research. Comprehensive assessment of climate change impacts requires a more accurate depiction of fine-grained climatic conditions and continuous long-term time coverage. Here, we present a series of improved 1-km Köppen-Geiger climate classification maps for six historical periods in 1979-2013 and four future periods in 2020-2099 under RCP2.6, 4.5, 6.0, and 8.5. The historical maps are derived from multiple downscaled observational datasets and the future maps are derived from an ensemble

of bias-corrected downscaled CMIP5 projections. In addition to climate classification maps, we calculate 12 bioclimatic variables at 1-km resolution, providing detailed descriptions of annual averages, seasonality, and stressful conditions of climates. The new maps offer higher classification accuracy than existing climate map products and demonstrate the ability to capture recent and future projected changes in spatial distributions of climate zones. On regional and continental scales, the new maps show accurate depictions of topographic features and correspond closely with vegetation distributions. We also

provide a heuristic application example to detect long-term global-scale area changes of climate zones. This high-resolution dataset of the Köppen-Geiger climate classification and bioclimatic variables can be used in conjunction with species distribution models to promote biodiversity conservation and to analyze and identify recent and future interannual or interdecadal changes in climate zones on a global or regional scale. The dataset referred to as KGClim, is publicly available at http://doi.org/10.5281/zenodo.5347837 for historical climate and http://doi.org/10.5281/zenodo.4542076 for future climate.

## 1 Introduction

Climate has direct impacts on the processes and functioning of the ecosystem as well as on the distribution of species. (Chen, Hill, Ohlemüller, Roy, & Thomas, 2011; Ordonez & Williams, 2013; Pinsky, Worm, Fogarty, Sarmiento, & Levin, 2013; Thuiller, Lavorel, Araújo, Sykes, & Prentice, 2005). The spatial patterns of climates have been often identified using the Köppen climate classification system (Köppen, 1931).

The Köppen classification system was designed to map the distribution of the world's biomes based on the amplitude and seasonal phase of annual cycles of surface air temperature and precipitation (Köppen, 1936). Compared with other human expertise based climate mapping methods (e.g., Holdridge, 1947; Thornthwaite, 1931; Walter & Elwood, 1975) and clustering approaches (e.g., Netzel & Stepinski, 2016), which suffer from a lack in meteorological basis, the Köppen classification demonstrates stronger correlation with distributions of biomes and soil types (Bockheim, Gennadiyev, Hammer, & Tandarich,

2005; Rohli, Joyner, Reynolds, & Ballinger, 2015). It provides an ecologically relevant and effective method to classify climate conditions by combining seasonal cycles of surface air temperature and precipitation (Cui, Liang, & Wang, 2021).

The Köppen classification has been widely applied in biological science, earth and planetary sciences, and environmental science (Rubel & Kottek, 2011). It is a convenient and integrated tool to identify spatial patterns of climate distributions and to examine relationships between climates and biological systems. It has been found useful for a variety of issues on climate

change, such as hydrological cycle studies (Peel, McMahon, Finlayson, & Watson, 2001), Arctic climate change (Feng et al., 2012; Wang & Overland, 2004), assessment of climate change impacts on ecosystem (Roderfeld et al., 2008), biome distribution (Rohli, Joyner et al., 2015) and biodiversity (Garcia, Cabeza, Rahbek, & Araújo, 2014).

There has been a resurgence in the application of the Köppen climate classification in climate change research over the recent decades (Cui, Liang, & Wang, 2021). The Köppen climate classification has been used to set up dynamic global vegetation

models (Poulter et al., 2011; Poulter et al., 2015), to characterize species composition (Brugger & Rubel, 2013), to model the species range distribution (Brugger & Rubel, 2013; Tererai & Wood, 2014; Webber et al., 2011), and to analyze the species growth behavior (Tarkan & Vilizzi, 2015). The Köppen classification has also been applied to detect the shifts in geographical distributions of climate zones (Belda, Holtanová, Kalvová, & Halenka, 2016; Chan & Wu, 2015; Feng et al., 2014; Mahlstein, Daniel, & Solomon, 2013). It also has the potential to aggregate climate information on warmth and precipitation seasonality

into ecologically important climate classes thereby simplifying spatial variability. This climate classification system adds a new direction to develop climate change metrics and can provide support for the growth of species distribution modelling (SDM).

The recent Köppen climate classification maps have a resolution ranging between 0.5° and 1-km (Cui, Liang, & Wang, 2021). Early published Köppen climate classification maps have a relatively low resolution of 0.5° (Belda, Holtanová, Halenka, &

Kalvová, 2014; Grieser, Gommes, Cofield, & Bernardi, 2006; Kottek, Grieser, Beck, Rudolf, & Rubel, 2006; Kriticos et al., 2012; Rubel & Kottek, 2010). Several map products used interpolation methods to obtain a higher resolution of ~0.1° (Kriticos et al., 2012; Peel, Finlayson, & McMahon, 2007; Rubel, Brugger, Haslinger, & Auer, 2017). Fine resolutions of at least 1-km are required to detect microrefugia and promote effective conservation. As the only 1-km global climate classification map product, Beck et al. (2018) provided global climate classification maps for two periods 1980-2016 and 2071-2100 under

RCP8.5. The maps were derived using climate data from WorldClim V1 and V2 (Fick & Hijmans, 2017; Booth et al., 2014), CHELSA V1.2 (Booth et al., 2014), and CHPclim V1 (Funk et al., 2015). To represent historical climates, they adjusted the

inconsistent temporal spans of climatology datasets to the period 1980-2016, by adding interpolated temperature change offsets or multiplying precipitation factors, which may lead to biased coverage of the historical period. Current classification accuracy, temporal coverage, spatial and temporal resolution of historical and future climate classification maps cannot sufficiently fulfil

the current needs of climate change research. Significant changes in the Köppen climates have been observed and projected in the recent two centuries (Belda et al., 2014; Chan & Wu, 2015; Chen & Chen, 2013; Rohli, Andrew, Reynolds, Shaw, & Vázquez, 2015; Yoo & Rohli, 2016). Previous studies found that large-scale shifts in climate zones have been observed over more than 5% of the total land area since the 1980s, and approximately 20.0% of the total land area is projected to experience climate zone changes under RCP8.5 by 2100 (Cui, Liang, & Wang, 2021). Detection of recent and future changes in climate

zones with the application of the Köppen climate maps needs more accurate depiction of fine-grained climatic conditions, continuous and longer temporal coverage.

This creates the urgent need for global maps of the Köppen climate classification with increased accuracy, finer spatial and temporal resolutions. Currently available global observational datasets of temperature and precipitation collected during the recent centuries, and the global climate simulations under alternative future climate scenarios have offered the possibility to

create a comprehensive dataset for past and future climates. In this study, we presented an improved long-term Köppen-Geiger climate classification map series for 1) six historical 30-yr periods of the observational record (1979-2008,1980-2009, 1981-2010, 1982-2011, 1983-2012, 1984-2013) and four future 30-yr periods (2020-2049, 2040-2069, 2060-2089, 2070-2099) under four RCPs (RCP2.6, 4.5, 6.0 and 8.5). To improve the classification accuracy and achieve a resolution as fine as 1-km (30 arc-second), we combined multiple datasets, including WorldClim V2 (Fick & Hijmans, 2017), CHELSA V1.2 (Booth et al.,

2014), CRU TS v4.03 (New, Hulme, & Jones, 2000), UDEL (Willmott & Matsuura, 2001), GPCC datasets (Beck, Grieser, & Rudolf, 2005) and bias-corrected downscaled Coupled Model Intercomparison Project Phase 5 (CMIP5) model simulations (Navarro-Racines, Tarapues, Thornton, Jarvis, & Ramirez-Villegas, 2020) (Table 1). We used the WorldClim Historical Climate Data V2 (Fick & Hijmans, 2017) to downscale the 0.5° climatology datasets including CRU, UDEL and GPCC, and derive high resolution climate data for the historical periods. To determine the final climate class, we used the climate class

with the highest agreement level from an ensemble of climate maps derived from different combinations of surface air temperature and precipitation products, as implemented in Beck et al. (2018). In addition to the Köppen-Geiger climate maps, we also calculated 12 bioclimatic variables at the same 1-km resolution using these climate datasets for the same historical and future periods. This dataset can be used to in conjunction with SDMs to promote biodiversity conservation, or to map plant functional type distributions for earth system model simulations, or to analyse and identify recent and future changes in climate

zones on a global or regional scale.

To validate the Köppen-Geiger climate classification maps, we used the station observations from Global Historical Climatology Network-Daily (GHCN-D) (Menne, Durre, Vose, Gleason, & Houston, 2012), and Global Summary Of the Day (GSOD) (National Climatic Data Center, NESDIS, NOAA, & U.S. Department of Commerce, 2015) database. At the regional and continental scale, we compared our Köppen-Geiger climate classification maps with previous map products, associated

maps of forest cover, and elevation distribution, for 1) regions with large spatial gradients in climates, including central and eastern Africa, Europe, North America, and 2) regions with sharp elevation gradients, including Tibetan Plateau, central Rocky Mountains, central Andes. Further, we conducted sensitivity analysis with respect to classification temporal scale, dataset input, and data integration methods. We also provided a heuristic example which used climate classification map series to detect the long-term area changes of climate zones, showing how the Köppen-Geiger climate classification map series can be

applied in climate change studies.

## 2 Datasets

**Table 1 Climatology datasets to generate present global maps of the Köppen climate classification with varied spatial resolutions**

| Dataset | Usage | Spatial Res. | Temporal Span | Variable | Source and Description |
|---|---|---|---|---|---|
| Present Köppen classification map series with resolution of 30 arc-second (1km) | | | | | |
| CRU | Map Input | 0.5° | 1979-2017 | T | Climatic Research Unit (CRU) TS v4.03 |
| UDEL | Map Input | 0.5° | 1979-2017 | T, P | U. of Delaware Precipitation and Air Temperature |
| WorldClim | Downscaling | 0.0083° | 1970-2000 | T, P | WorldClim Historical Climate Data V2 |
| CHELSA | Map Input | 0.0083° | 1979-2013 | T, P | Climatologies at high resolution for the earth's land surface areas (CHELSA) |
| GPCC | Map Input | 0.5° | 1979-2016 | P | Global Precipitation Climatology Centre (GPCC) |
| PREC/L | Data Selection | 0.5° | 1979-2012 | P | NOAA's PRECipitation REConstruction over Land (PREC/L) |
| GHCN_CAMS | Data Selection | 0.5° | 1979-2017 | T | GHCN_CAMS Gridded 2m Temperature (Land) |
| Future Köppen classification map series with resolution of 30 arc-second (1km) | | | | | |
| CMIP5 | Map Input | 0.0083° | 2020-2100 | T, P | CCAFS-Climate Statistically Downscaled Delta Method CMIP5 data |
| WorldClim | Downscaling | 0.0083° | 1970-2000 | T, P | WorldClim Historical Climate Data V2 |

Table 1 lists the climatology datasets with global coverage and on a monthly time step, used to generate historical and future Köppen-Geiger climate map series. The present 1-km Köppen-Geiger classification map series for 1979-2013 was derived from the Climatologies at High-resolution for the Earth's Land Surface Areas (CHELSA) V1.2 (Karger et al., 2017),

WorldClim Historical Climate Data V2 (Fick & Hijmans, 2017) and the statistically downscaled Climatic Research Unit (CRU) TS v4.03 (New et al., 2000), University of Delaware Precipitation and Air Temperature (UDEL) (Willmott & Matsuura, 2001) and Global Precipitation Climatology Centre (GPCC) (Beck et al., 2005) datasets. To decide the datasets to use, we conducted a sensitivity analysis on the input climatology datasets and utilized monthly air temperature datasets from CRU, UDEL, GHCN_CAMS Gridded 2m Temperature (Fan & Dool, 2008) and monthly precipitation datasets from GPCC, UDEL,

NOAA's PRECipitation REConstruction over Land (PREC/L) (Chen, Xie, Janowiak, & Arkin, 2002). Evaluation results indicated that incorporating only CRU, UDEL temperature datasets and UDEL, GPCC precipitation datasets and excluding GHCN_CAMS and PREC/L datasets led to higher accuracy in the classification results. Therefore, we chose CRU, UDEL, and GPCC datasets as the classification system input to boost the final accuracy.

To explicitly correct topographic effect, we used 1-km CHELSA V1.2 and WorldClim V2 datasets in addition to the 0.5° resolution datasets. The CHELSEA dataset statistically downscaled temperature data from the ERA-Interim climatic reanalysis. For precipitation data, it incorporated multiple orographic predictors and performed bias correction (Karger et al., 2017). With major topo-climatic drivers considered, the CHELSA dataset demonstrated good performance in ecological science studies. CHELSA data exhibited comparable accuracy for temperatures and better predicted precipitation patterns based on the validation results (Karger et al., 2017).

We produced the future Köppen classification map series using the CCAFS climate statistically bias-corrected and downscaled CMIP5 projections (Navarro-Racines et al., 2020). The CCAFS presented a global database of future climates developed by a climate model bias correction method based on the CMIP5 GCM simulations (Taylor, Stouffer, & Meehl, 2012) archive, coordinated by the World Climate Research Programme in support of the IPCC Fifth Assessment Report (AR5) (Hartmann et al., 2013). The total is 35 GCMs, and all RCPs, RCP 2.6, 4.5, 6.0 and 8.5 (Table S1). Projections are available at varied coarse scales (70–400km). To achieve high-resolution (1km) climate representations, downscaling method has been applied with the use of the WorldClim data (Fick & Hijmans, 2017) . Technical evaluation showed that the bias-correction method that CCAFS data applied reduced climate model bias by 50–70%, which could potentially address the bias issue in model simulations for the threshold-based Köppen classification scheme (Navarro-Racines et al., 2020).

## 3 Methodology

### 3.1 Köppen-Geiger climate classification

**Table 2 Criteria of the Köppen-Geiger climate classification with temperature in oC and precipitation in mm.**

| 1st | 2nd | 3rd | Description | Criterion |
|---|---|---|---|---|
| A | | | Tropical | Not (B) & $T_{cold} \geq 18$ |
| | f | | - Rainforest | $P_{dry} \geq 60$ |
| | m | | - Monsoon | Not (Af) & $P_{dry} \geq 100$-$MAP/25$ |
| | w | | - Savannah | Not (Af) & $P_{dry} < 100$-$MAP/25$ |
| B | | | Arid | $MAP < 10 \times P_{threshold}$ |
| | W | | - Desert | $MAP < 5 \times P_{threshold}$ |
| | S | | - Steppe | $MAP \geq 5 \times P_{threshold}$ |
| | | h | -- Hot | $MAT \geq 18$ |
| | | k | -- Cold | $MAT < 18$ |
| C | | | Temperate | Not (B) & $T_{hot} > 10$ & $-3 < T_{cold} < 18$ |
| | w | | - Dry winter | $P_{wdry} < P_{swet}/10$ |
| | s | | - Dry summer | $Not (w)$ & $P_{sdry} < 40$ & $P_{sdry} < P_{wwet}/3$ |
| | f | | - Without dry season | Not (s) or (w) |

| | | | |
|---|---|---|---|
| | a | -- Hot summer | $T_{hot} \geq 22$ |
| | b | -- Warm summer | Not (a) & $T_{mon10} \geq 4$ |
| | c | -- Cold summer | Not (a or b) & $1 \leq T_{mon10} < 4$ |
| D | | Boreal | Not (B) & $T_{hot} > 10$ & $T_{cold} \leq -3$ |
| | w | - Dry winter | $P_{wdry} < P_{swet}/10$ |
| | s | - Dry summer | Not (w) & $P_{sdry} < 40$ & $P_{sdry} < P_{wwet}/3$ |
| | f | - Without dry season | Not (s) or (w) |
| | a | - Hot summer | $T_{hot} \geq 22$ |
| | b | - Warm summer | Not (a) & $T_{mon10} \geq 4$ |
| | c | - Cold summer | Not (a), (b) or (d) |
| | d | - Very cold winter | Not (a) or (b) & $T_{cold} < -38$ |
| E | | Polar | Not (B) & $T_{hot} \leq 10$ |
| | T | - Tundra | $T_{hot} > 0$ |
| | F | - Frost | $T_{hot} \leq 0$ |

$MAT$ = mean annual air temperature (°C); $T_{cold}$ = the air temperature of the coldest month (°C); $T_{hot}$ = the air temperature of the warmest month (°C); $T_{mon10}$ = the number of months with air temperature >10 °C; $MAP$ = mean annual precipitation (mm y$^{-1}$); $P_{dry}$ = precipitation in the driest month (mm month$^{-1}$); $P_{sdry}$ = precipitation in the driest month in summer (mm month$^{-1}$); $P_{wdry}$ = precipitation in the driest month in winter (mm month$^{-1}$); $P_{swet}$ = precipitation in the wettest month in summer (mm month$^{-1}$); $P_{wwet}$ = precipitation in the wettest month in winter (mm month$^{-1}$); $P_{threshold}$=2×$MAT$ if >70% of precipitation falls in winter, $P_{threshold}$=2×$MAT$+28 if >70% of precipitation falls in summer, otherwise $P_{threshold}$=2×MAT+14.

The Köppen climate classification scheme was first introduced by Wladimir Köppen (1884). It is one of the earliest quantitative classification systems of Earth's climates. Its modification, the Köppen-Geiger classification (KGC) was first published in 1936 (Köppen, 1936), developed by Wladimir Köppen and Rudolf Geiger. KGC identifies climates based on their effects on plant growth from the aspects of warmth and aridity, and classifies climate into five main climate classes and 30 subtypes (Rubel & Kottek, 2011). The five main climate zones distinguish between plants of the tropical climate zone (A), the arid climate zone (B), the temperate climate zone (C), the boreal climate zone (D) and the polar climate zone (E), referring to the five major climate zones (Sanderson, 1999). All these main climate zones are thermal zones except the arid (B) climate zone, which is defined based on precipitation threshold.

This research followed the Köppen-Geiger climate classification as described in Kottek et al. (2006), and Rubel & Kottek (2010). This latest version of the KGC scheme was first presented by Geiger (1961) (Table 2). Several existing Köppen-Geiger climate map products, including Peel et al. (2007), Kriticos et al. (2012), and Beck et al. (2018) applied the KGC scheme modified following Russell (1931). Russell (1931) adjusted the definition of the boundary of temperate (C) and boreal (D) climate zones using the coldest monthly temperature > 0 °C instead of >-3 °C. This threshold was proposed because the 0°C line fits the distribution of the topographical features and vegetation in western United States, where at that time meteorological stations were sparsely distributed (Jones, 1932). However, the application of 0°C boundary to the global climates has not been

validated. Therefore, this research didn't utilize the Russell's modification (1931) and followed the latest version KGC proposed by Geiger (1961).

## 3.2 Statistical downscaling

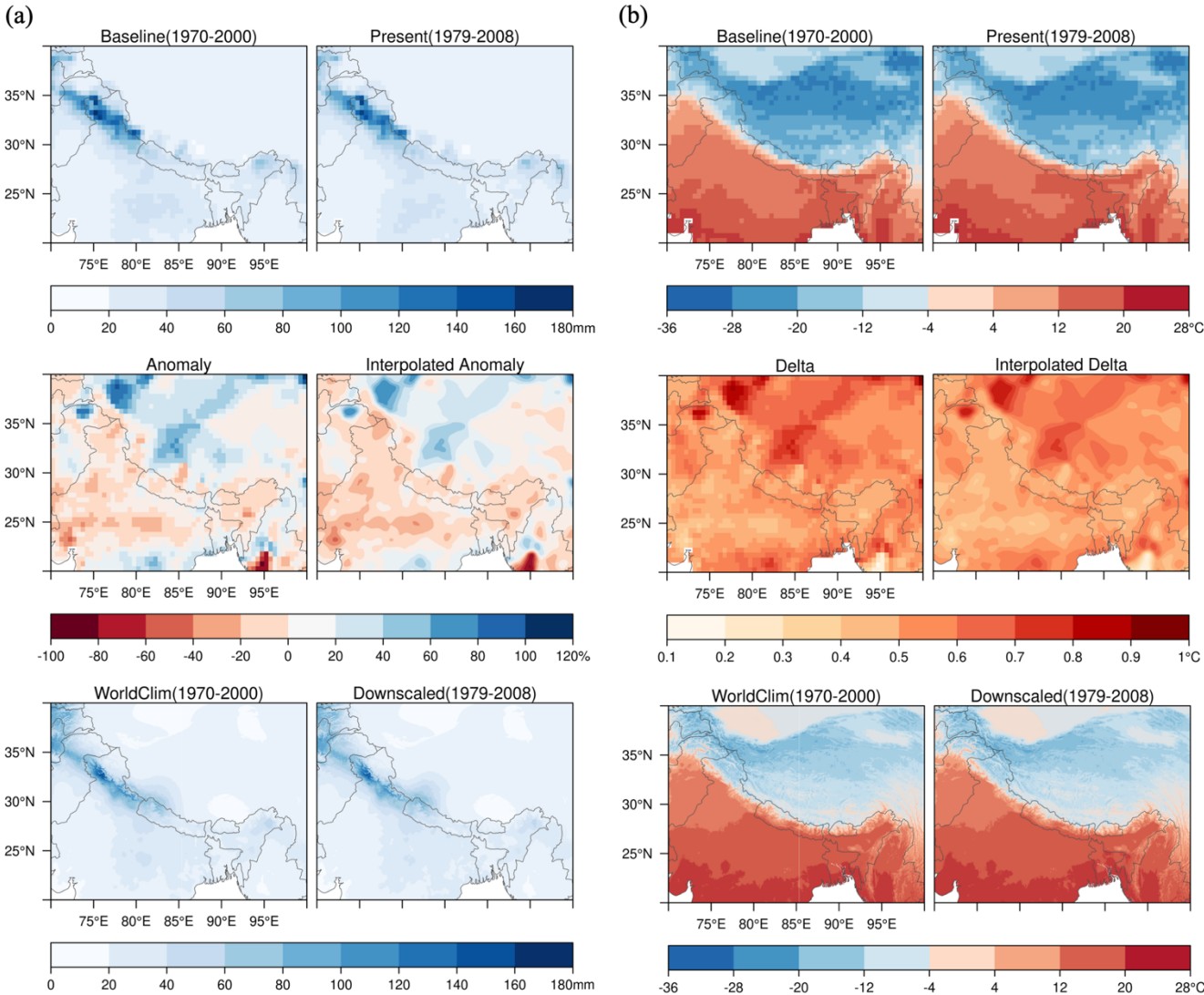

**Figure 1. Illustration of the downscaling process.** (a) Anomaly downscaling method with January total precipitation from GPCC dataset and (b) delta downscaling method with January temperature from CRU dataset. Baseline (1970-2000) and present-day climate data (e.g. 1979-2008) are from CRU, UDEL, or GPCC datasets, which have a coarse spatial resolution of 0.5°. Precipitation anomaly is change factor of monthly precipitation from baseline to present-day climates. Temperature delta is change in monthly air temperature from baseline to present-day climates. WorldClim (1970-2000) climate data is adjusted by multiplying 30 arc-second interpolated anomaly (for precipitation) or adding 30 arc-second interpolated delta (for temperature) to generate the downscaled climate surfaces with 30 arc-second resolution. Precipitation values in mm/month and temperature values in °C.

Due to limited number of available observational datasets with high resolution and long-term continuous temporal coverage, the research implemented the delta method by applying a delta change or change factor (Hay, Wilby, & Leavesley, 2000; Wilby & Wigley, 1997) onto the WorldClim historical observations (Fick & Hijmans, 2017) to achieve 30-yr average climatology data with a 1-km resolution based on the CRU, UDEL and GPCC datasets. The delta method is a statistical downscaling method that assumes that the relationship between climatic variables remain relatively constant at local scale (Wilby & Wigley, 1997). We applied delta method to downscale the long-term (30-yr) mean climates using coarse-resolution monthly climatology datasets. The delta changes or change factors are calculated as the differences between the 30-yr long-term means of temperature or precipitation of baseline (1970-2000) and present-day climates. The delta method comprises the following four steps: 1) calculate 30-yr averages for baseline (1970-2000) and present day of monthly temperature and precipitation; 2) calculate anomaly for precipitation and delta for temperature; 3) apply thin-plate splines interpolation (TPS) to create 1km surface of precipitation anomaly and temperature delta; 4) multiply anomaly or add delta to historical climates based on WorldClim dataset (Fig. 1).

First, using monthly time series from CRU, UDEL and GPCC datasets, we calculated 30-yr means as a baseline (1970-2000), for each climatology dataset and each variable. We used 1970-2000 as baseline period, for consistency with WorldClim Historical Climate Data V2. Next, we calculated 30-yr means for each month and each 30-yr present-day period in 1979-2013 We then calculated anomalies as proportional differences between present-day and baseline in total precipitation and delta as difference in temperature. To derive 30 arc-second (1-km) anomaly or delta surfaces, we applied thin-plate splines (TPS) interpolation (Craven & Wahba, 1978; Franke, 1982; Schempp, Zeller, & Duchon, 1977) on precipitation anomaly and temperature delta. TPS has been widely used in climate science (Hijmans, Cameron, Parra, Jones, & Jarvis, 2005; Navarro-Racines et al., 2020) as it produced a smooth and continuous surface, which is infinitely differentiable. Last, we multiplied the change factor or added the delta to the WorldClim (1970-2000) data to get downscaled present-day monthly climate data.

Our future Köppen-Geiger map series are based on an ensemble of maps derived from the CCAFS bias-corrected and downscaled climate projections, which include 35 CMIP5 GCMs, and 4 RCPs (Navarro-Racines et al., 2020). Large misclassifications exist within the GCMs as detected in previous assessment of large areas ranging between 20-50% of the total land area (Cui, Liang, & Wang, 2021). Deficiencies in model physics are also more likely to contribute to uncertainties in the maps than grid size or reference dataset limitations (Tapiador, Moreno, & Navarro, 2019). Multi-model mean and delta-change method can mitigate the bias effects from the threshold-based classification scheme and have been utilized to simulate better results of climate classification (Hanf, Körper, Spangehl, & Cubasch, 2012). Therefore, we chose the CCAFS bias-corrected and downscaled CMIP5 projections (Navarro-Racines et al., 2020) to reduce the amplified errors due to uncertainty of climate projections. Navarro-Racines et al. (2020) interpolated anomalies of original GCM outputs using thin plate spline spatial interpolation to achieve a baseline climate with a 1km surface. Then they applied delta method to the interpolated baseline climates to correct the model biases (Hay et al., 2000; Ho, Stephenson, Collins, Ferro, & Brown, 2012).

## 3.3 Data Integration

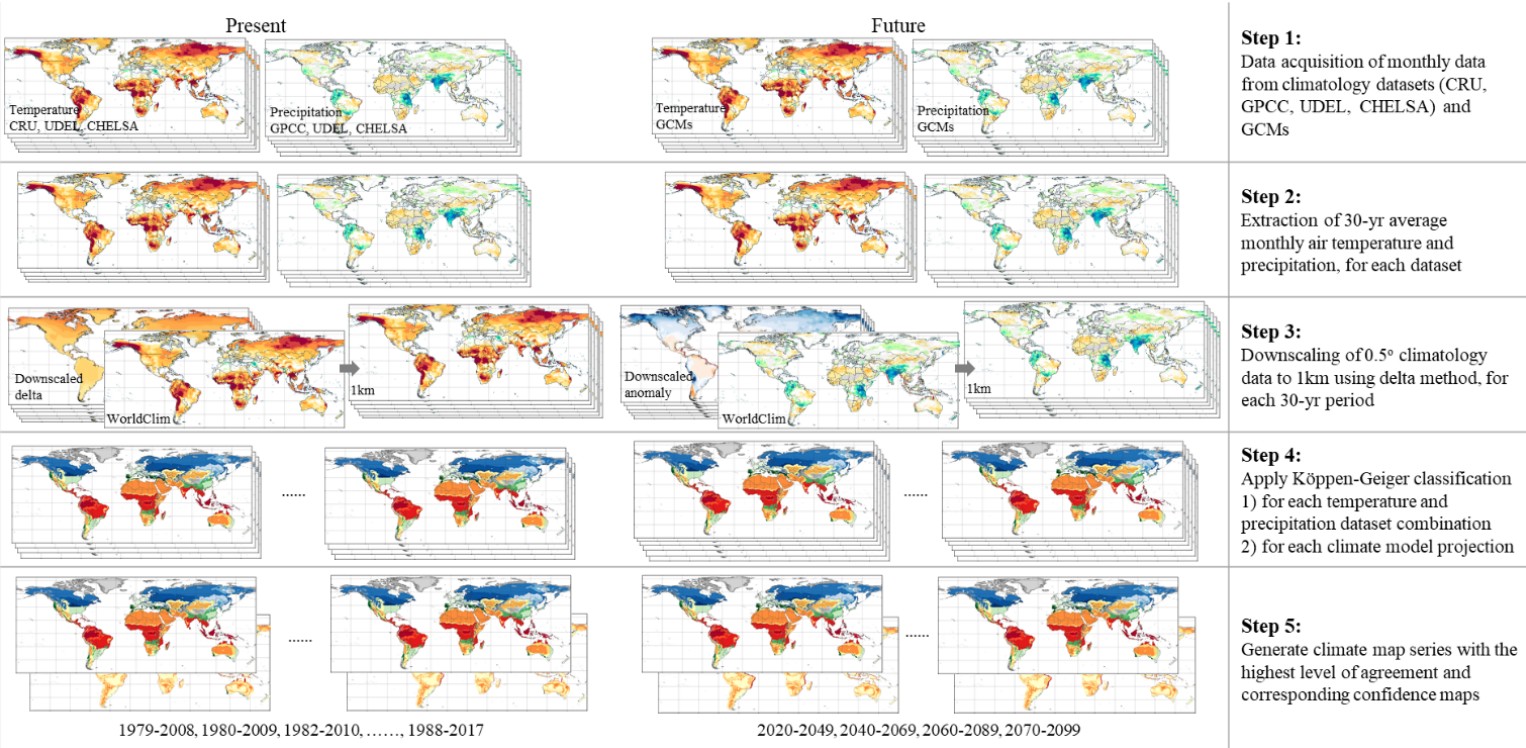

**Figure 2. Step by step process to generate Köppen-Geiger climate map series.**

The historical Köppen-Geiger climate classification map series was generated using the highest confidence class from an ensemble of maps using all combinations of surface air temperature and precipitation products (Fig. 2), as described in Beck et al. (2018). The highest confidence was given to the most common climate class for each grid cell. The final historical climate map series were derived using the climate class with the highest level of confidence in an ensemble of 3 × 3 = 9 classification maps based on combinations of the 3 precipitation datasets (CRU, UDEL, and CHELSA) and 3 surface air temperature datasets (GPCC, UDEL, and CHELSA). To further test the sensitivity of the method using the climate with the highest level of agreement, we incorporated another data integration method using the mean of multiple datasets. We quantified the degree of confidence placed in the Köppen-Geiger climate map series using the degree of confidence at the grid cell level calculated by dividing the occurrence frequency of the climate class with the highest level of agreement by the ensemble size. The calculated confidence level can be viewed as the agreement degree in classification resulted derived from different climatology datasets.

The future Köppen-Geiger climate classification map series under 4 RCPs, were derived based on the most common climate class from an ensemble of future climate maps. We generated a future Köppen-Geiger climate classification map for each climate model projection, using the CCAFS bias-corrected and downscaled CMIP5 GCM dataset. For example, the future

Köppen-Geiger climate classification map series under RCP8.5 was derived from an ensemble of 30 maps based on 30 CMIP5 models. The level of confidence was estimated using the ratio between the frequency of the climate class with the highest level of agreement in the future map results, and the ensemble size.

## 3.4 Validation

    We validated the historical climate maps using the station observations from Global Historical Climatology Network-Daily
(GHCN-D) (Menne et al., 2012) and Global Summary Of the Day (GSOD) database (National Climatic Data Center et al., 2015) as reference data. GHCN-D dataset provides daily climate data over global land areas and contains records from over 80,000 weather stations worldwide, about one third of which have both temperature and precipitation data available (Menne et al., 2012). GSOD dataset includes global daily summary data over 9,000 stations, of which the historical data from 1973 being the most complete (National Climatic Data Center et al., 2015). For each station, time series of monthly temperature and
precipitation were calculated from the daily observations with months with <15 daily values discarded. Then if ≥6 months are present, monthly climatology were generated subsequently by averaging the monthly means for the given 30-yr period. We removed duplicate stations in the two datasets and discarded stations with gap years or missing data in the given 30 years. For each station and each 30-yr period, we applied the Köppen-Geiger climate classification, and then evaluated overall classification performance for each climate map using total accuracy, which is defined as the percentage of correct classes,
and average precision, which is averaged fraction of correct classification for all climate classes.

    Using the same validation datasets and station selection process, we also evaluated the previous climate maps from Beck et al., (2018) Kriticos et al., (2012), Peel et al., (2007), and Kottek et al., (2006). We applied the same Köppen-Geiger climate classification criteria described in the preivous studies to assess the overall accuracy of the map products. To further validate the climate classification results, we performed sensitivity analysis on the data integration method, the climate classificaiton
time scale, and climatology dataset input, using the same validation datasets from GHCN-D and GSOD. In addition, we compared the climate classification results with forest cover and elevation maps, and with the two high-resolution comparable climate map products, Beck et al., (2018) (1-km) and Kriticos et al., (2012) (0.167 º), at regional and continental scale. The forest cover map we used is the 2000 30m Landsat-based forest cover map (Hansen et al., 2013). The elevation data is from the NASA SRTM Digital Elevation 30m data (Farr et al., 2007).

**4 Results and Discussion**

## 4.1 Historical Köppen-Geiger climate maps

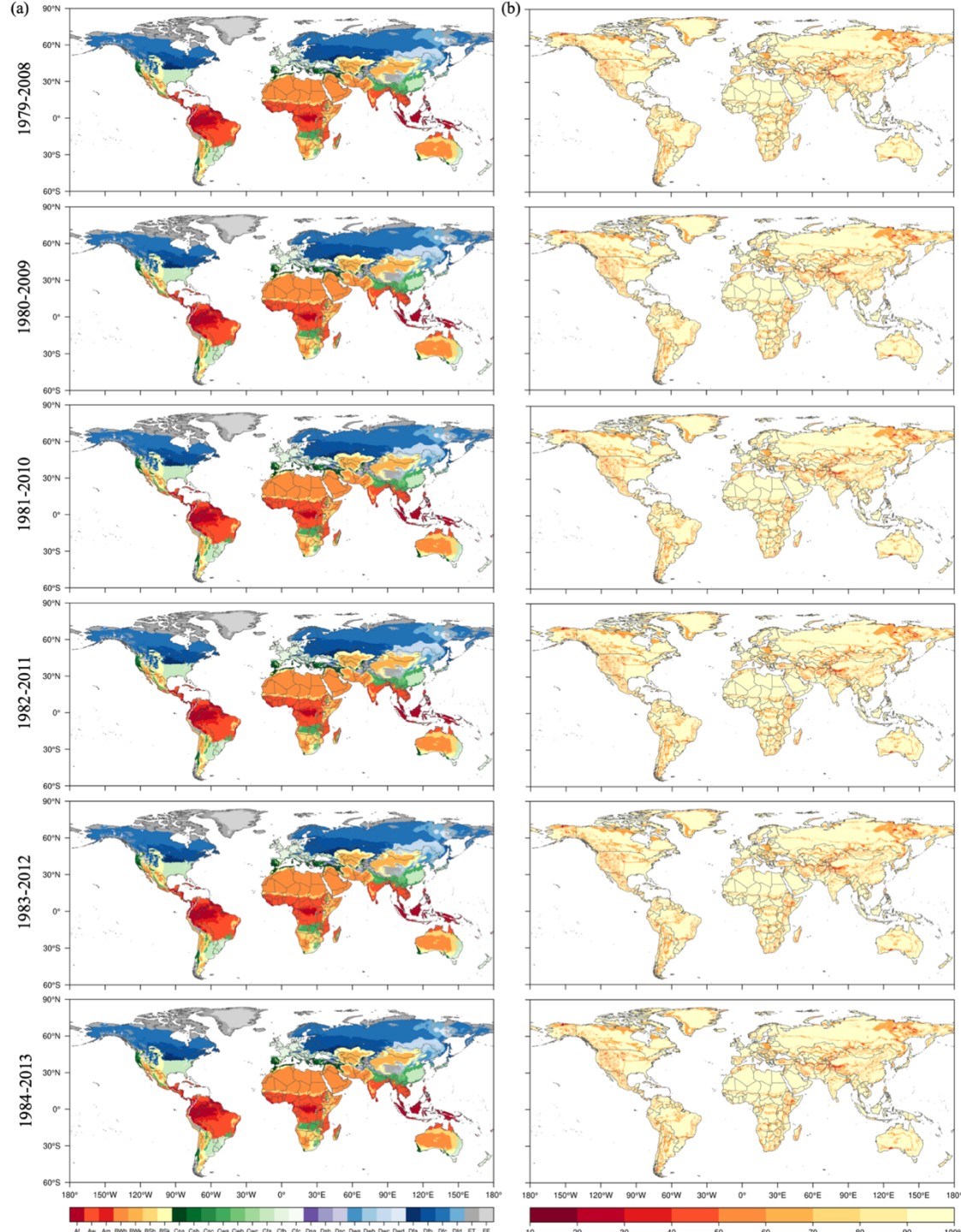

**Figure 3 Global maps of the Köppen-Geiger climate classification for the historical periods (1979-2008, 1980-2009, 1981-2010, 1982-1011, 1983-2012, 1984-2013) and associated classification confidence levels.** (a) Historical maps of the Köppen-Geiger climate classification and (b) confidence levels associated with the Köppen-Geiger climate classification.

Global map series of the Köppen-Geiger climate classification for historical periods and associated corresponding confidence levels are shown in Figure 3. Based on the distribution of confidence level, over 90% of the land area exhibit high level of confidence as classification results based on different climate data show excellent agreement. Relatively lower confidence

level and large discrepancy in classification results are found especially in mountainous regions such as Andes Mountains, Rocky Mountains, Tibetan Plateau, and major climate transitional zones located in mid and high latitudes of Northern Hemisphere, Central Africa, and Central Asia.

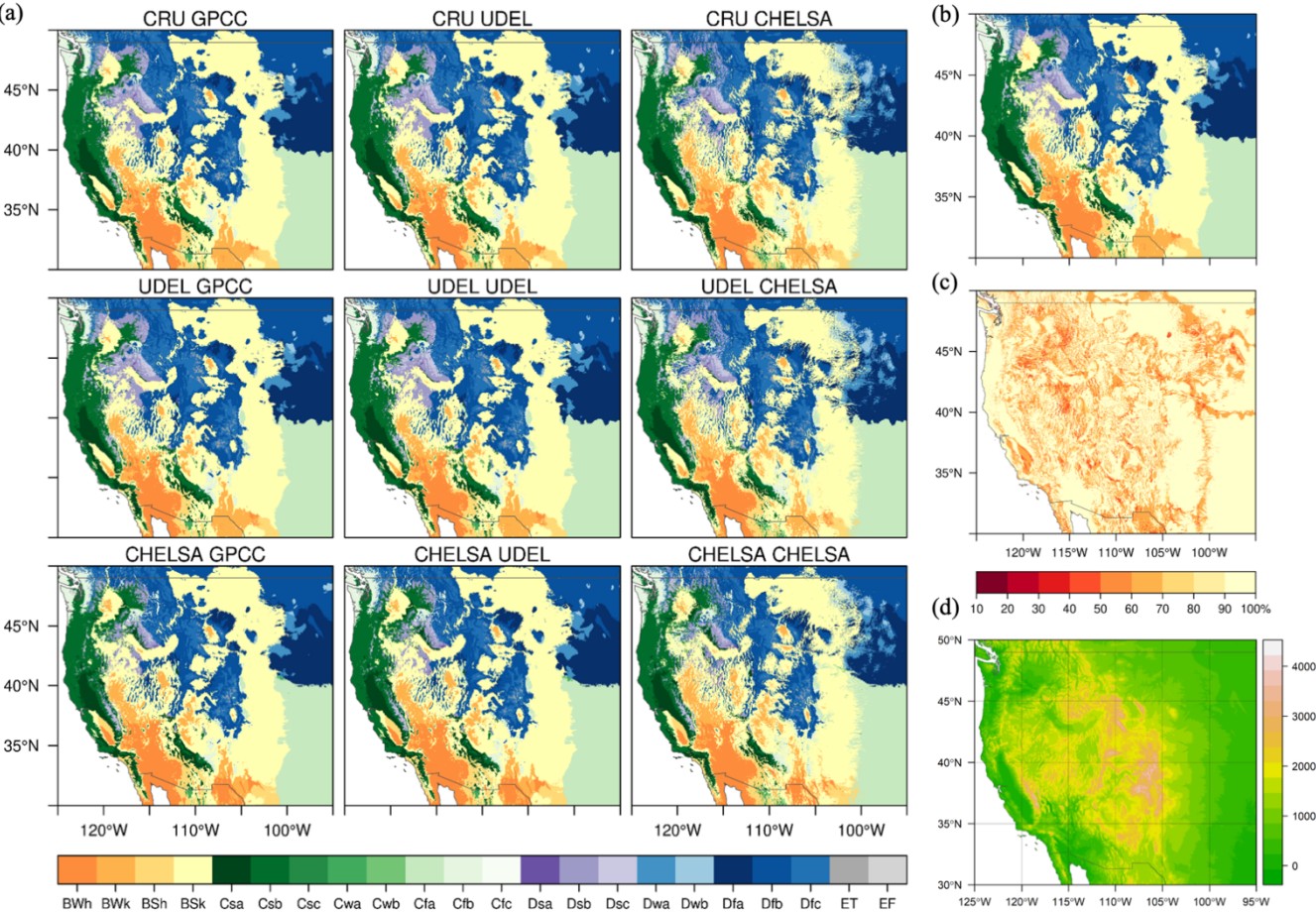

**Figure 4. Present Köppen-Geiger classification and confidence map for 1979-2008 with resolution of 1km for the central Rocky**
**Mountains in North America.** (a) Climate maps based on the 9 combinations of the 3 precipitation datasets × 3 surface air temperature datasets, (b) the final climate map derived from the most common climate class among the 9 climate maps, (c) confidence level distribution of the final climate map, and (d) elevation map for the the central Rocky Mountains in North America.

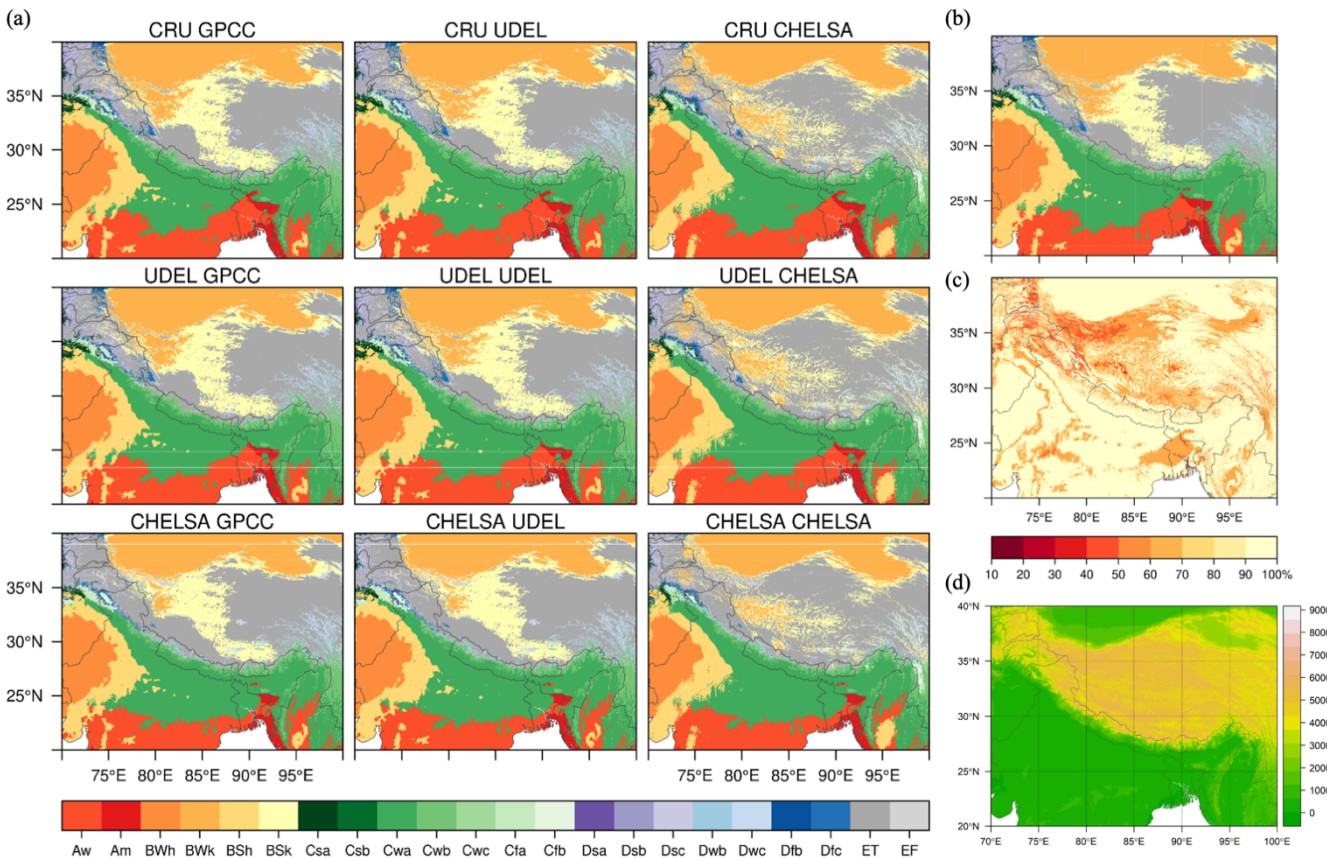

**Figure 5. Present Köppen-Geiger classification and confidence map for 1979-2008 with resolution of 1km for the Tibetan Plateau.**
(a) Climate maps based on the 9 combinations of the 3 precipitation datasets × 3 surface air temperature datasets, (b) the final climate map derived from the most common climate class among the 9 climate maps, (c) confidence level distribution of the final climate map, and (d) elevation map for the Tibetan Plateau.

Regional distributions of climatic conditions are largely created by local variation in topography in rugged terrain (Dobrowski et al., 2013; Franklin et al., 2013). The climate classification and confidence level maps of mountainous areas of Central Rocky Mountains and Tibetan Plateau are shown in Figure 4 and 5 respectively. For each combination of precipitation and surface air temperature datasets, we generated a Köppen-Geiger climate classification map (see Fig. 4a and 5a for 1979-2008 maps for the central Rocky Mountains and Tibetan Plateau). The final Köppen-Geiger classification map is derived based on the most common climate type among all the climate maps (Fig. 4b and 5b). We then calculated corresponding confidence levels to quantify the uncertainty in the classification maps (Fig. 4c and 5c). The uncertainty in climate classification in mountainous areas is attributed to the uncertainty existing in climate data, especially precipitation data. In rugged terrain, CHELSA precipitation data shows more detailed precipitation patterns, causing disagreement in classificaion results of the 3[rd] level climate classes which depict precipitation seasonality.

## 4.2 Validation

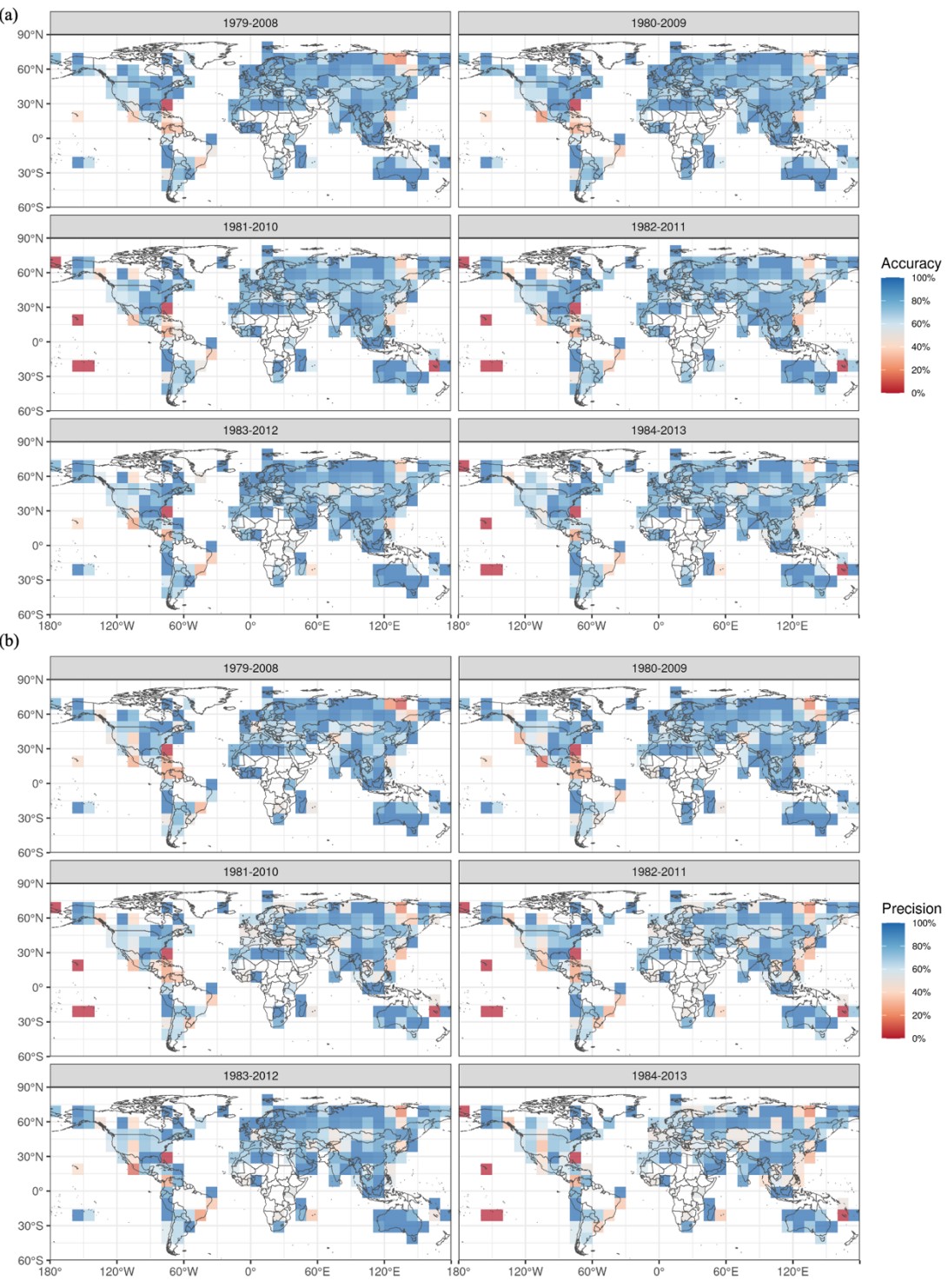

**Figure 6. Validation of the historical Köppen-Geiger climate map series (1979-2008, 1980-2009, 1981-2010, 1982-2011, 1983-2012, 1984-2013).** (a) Small-scale accuracy of historical Köppen-Geiger climate maps. (b) Small-scale precision of historical Köppen-Geiger climate maps. Climate classificaiton has been applied for each station. The small-scale accuracy and precision are calculated based on the classification results of all the stations within the given region, with a minimum of 3 stations in the 5° search radius.

We validated the historical climate maps using the station observations from Global Historical Climatology Network-Daily (GHCN-D) (Menne et al., 2012) and Global Summary Of the Day (GSOD) database (National Climatic Data Center et al., 2015). Figure 6 shows the small-scale distributions of total accuracy and average precision for historical Köppen-Geiger climate map series with 10° grid cells. Due to uneven distributions of weather stations, remote areas in the Pacific islands, Central Africa, and Amazon Forest suffer from a lack of station observations or an underrepresented validation results.

**Table 3 Continental and global overall accuracy, average precision, and confidence level of the historical Köppen-Geiger climate map series (1979-2008, 1980-2009, 1981-2010, 1982-2011, 1983-2012, 1984-2013).** The overall accuracy is calculated as the percentage of correct climate classes using ground observations, and average precision is averaged fraction of correct classification for all climate classes. Confidence level values shows the 95% confidence interval of the confidence level for each continent, and the whole globe. All the values are presented in percentage.

| | Region | Africa | Asia | Oceania | Europe | North America | South America | Global |
|---|---|---|---|---|---|---|---|---|
| Accuracy | 1979-2008 | 88.24% | 84.05% | 92.39% | 85.11% | 79.37% | 69.18% | 83.25% |
| | 1980-2009 | 87.67% | 85.00% | 90.11% | 84.24% | 76.94% | 70.00% | 82.96% |
| | 1981-2010 | 85.71% | 84.29% | 93.48% | 84.23% | 75.61% | 68.75% | 82.63% |
| | 1982-2011 | 83.78% | 85.06% | 91.30% | 84.10% | 74.79% | 68.90% | 82.42% |
| | 1983-2012 | 85.43% | 83.64% | 92.39% | 83.51% | 71.99% | 66.67% | 81.48% |
| | 1984-2013 | 85.81% | 81.32% | 92.39% | 84.38% | 71.84% | 68.00% | 81.62% |
| | Average | 86.11% | 83.89% | 92.01% | 84.26% | 75.09% | 68.58% | 82.39% |
| Precision | 1979-2008 | 80.24% | 72.77% | 92.77% | 75.71% | 64.41% | 66.20% | 71.27% |
| | 1980-2009 | 88.33% | 73.40% | 89.83% | 75.58% | 65.15% | 68.11% | 73.39% |
| | 1981-2010 | 79.54% | 71.19% | 94.21% | 74.77% | 67.75% | 67.63% | 74.10% |
| | 1982-2011 | 70.42% | 71.34% | 91.37% | 75.61% | 70.62% | 66.65% | 74.24% |
| | 1983-2012 | 71.54% | 68.99% | 92.67% | 69.82% | 66.73% | 64.33% | 72.41% |
| | 1984-2013 | 71.66% | 68.08% | 92.55% | 76.30% | 67.95% | 65.17% | 74.59% |
| | Average | 76.96% | 70.96% | 92.23% | 74.63% | 67.10% | 66.35% | 73.33% |
| Confidence level | 1979-2008 | 94.93±0.002% | 92.08±0.002% | 91.82±0.002% | 92.29±0.002% | 94.55±0.004% | 92.31±0.003% | 92.94±0.002% |
| | 1980-2009 | 94.91±0.002% | 92.14±0.002% | 91.73±0.002% | 92.39±0.002% | 94.65±0.004% | 92.24±0.003% | 92.95±0.002% |
| | 1981-2010 | 94.89±0.002% | 92.17±0.002% | 91.63±0.002% | 92.43±0.002% | 94.51±0.004% | 92.18±0.003% | 92.92±0.002% |
| | 1982-2011 | 94.92±0.002% | 92.16±0.002% | 91.48±0.002% | 92.41±0.002% | 94.35±0.004% | 92.13±0.003% | 92.87±0.002% |
| | 1983-2012 | 94.96±0.002% | 92.16±0.002% | 91.31±0.002% | 92.54±0.002% | 94.37±0.004% | 92.05±0.003% | 92.87±0.002% |
| | 1984-2013 | 94.97±0.002% | 91.22±0.002% | 91.32±0.002% | 92.52±0.002% | 94.45±0.004% | 92.00±0.003% | 92.87±0.002% |
| | Average | 94.93±0.002% | 91.99±0.002% | 91.55±0.002% | 92.43±0.002% | 94.48±0.004% | 92.15±0.003% | 92.90±0.002% |


We summarized the overall accuracy, average precision, and confidence levels for each continent and the whole globe (Table 3). The global overall classification accuracy of the historical Köppen-Geiger climate maps is estimated to be 82.39% with the lowest in South America (68.58%) and highest in Oceania (92.01%). The global average precision, which is calculated as averaged fraction of correct classification for all climate classes, is 73.33%. Similar to overall accuracy, South American has

the lowest precision level, equal to 66.35% and Oceania the highest, 92.23%. Having a good correspondence with accuracy and precision values, the continental average confidence levels range from 91.55% to 94.93%, and the global level is 92.90% (Table S2). Overall, the spatial patterns of total accuracy and average precision show good correspondence with classification confidence levels (Figure 3), indicating a potential of confidence level to represent classification uncertainty.

**Table 4 Accuracy of the 1km Köppen-Geiger climate map series derived from different combinations of temperature and precipitation dataset input, and by different means of integration of multiple datasets.** The values represent overall accuracy based on the technical validation using ground observation as reference.

| Temperature | CHELSA, Downscaled CRU and UDEL | | Downscaled CRU and UDEL | | CHELSA |
| --- | --- | --- | --- | --- | --- |
| Precipitation | CHELSA, Downscaled GPCC and UDEL | | Downscaled GPCC and UDEL | | CHELSA |
| Integration of multiple datasets | Highest agreement level | Mean of multiple datasets | Highest agreement level | Mean of multiple datasets | - |
| 1979-2008 | 83.25% | 83.66% | 83.13% | 83.33% | 79.72% |
| 1980-2009 | 82.96% | 83.44% | 82.74% | 82.78% | 79.14% |
| 1981-2010 | 82.63% | 82.86% | 81.95% | 82.38% | 78.03% |
| 1982-2011 | 82.42% | 82.73% | 81.93% | 82.11% | 78.47% |
| 1983-2012 | 81.48% | 82.34% | 81.14% | 81.49% | 78.32% |
| 1984-2013 | 81.62% | 82.05% | 80.84% | 81.27% | 78.26% |
| 1985-2014 | - | - | 80.23% | 80.86% | - |
| 1986-2015 | - | - | 79.79% | 80.58% | - |
| 1987-2016 | - | - | 78.76% | 79.62% | - |
| 1988-2017 | - | - | - | 78.65% | - |
| Average | 82.39% | 82.85% | 81.17% | 81.31% | 78.66% |
| 1980-2017 (Beck et al. 2018) | 77.65% | | | | |
| 1961-1990 (Kriticos et al., 2012) | 64.70% | | | | |

Using the same validation datasets from GHCN-D and GSOD, we tested sensitivity of the climate map series using different

combinations of temperature and precipitation dataset, and different method of data integration (Table 3). Results indicated an average total accuracy of the 1km Köppen-Geiger classification maps generated with all the CHELSA, donwscaled CRU, GPCC and UDEL datasets and with only downscaled CRU, GPCC, UDEL datasets as 82.39% and 81.17% respecively. Using the mean of multiple datasets which can potentially reduce the data bias, led to better classification results. We

estiomtated the total accuracy of the previous high resolution Köppen-Geiger climate map products using the same

validation datasets. We applied the same classification system described in the previous studies and the same time period of the previous climate map product to process the station observation data and estimate their overall accuracy. Compared with the previous high resolution Köppen-Geiger climate map products, Beck et al. (2018) and Kriticos et al., (2012), the newly generated Köppen-Geiger climate map series showed greater accuracy in total.

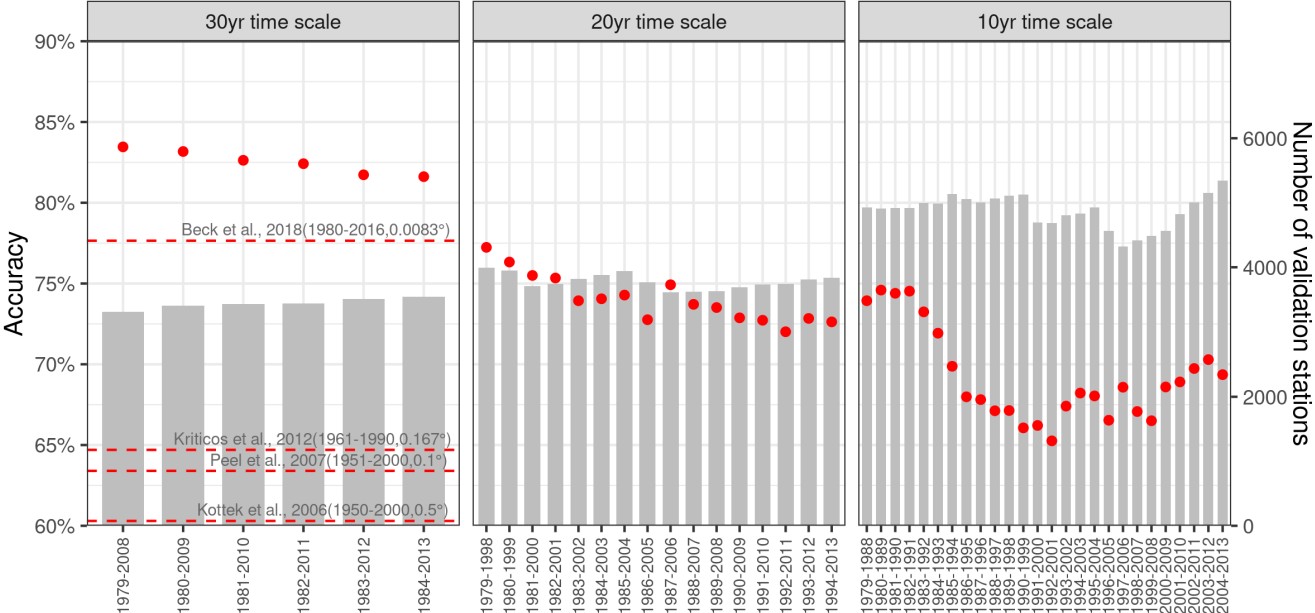

**Figure 7. Validation of downscaled data of bioclimatic variables and the generated Köppen-Geiger climate map.**

We conducted sensitivity analysis of the Köppen classification scheme and tested multiple time scales, 10-yr, 20-yr, and 30-yr. The selection criteria of station observations were adjusted accordingly based on the time scale utilized. Accuracy results exhibited decreasing accuracy for shorter time scale (Fig. 7). Further, we estimated the total accuracy for the Köppen-Geiger climate classification maps from previous studies, Beck et al., (2018) Kriticos et al., (2012), Peel et al., (2007), and Kottek et

al., (2006), using the same validation dataset and consistent Köppen-Geiger climate classification scheme the corresponding study applied. The validation results demonstrate that the new Köppen-Geiger maps have comparatively higher overall accuracy than all the previous studies.

**4.3 Regional and continental scale comparison**

(a)

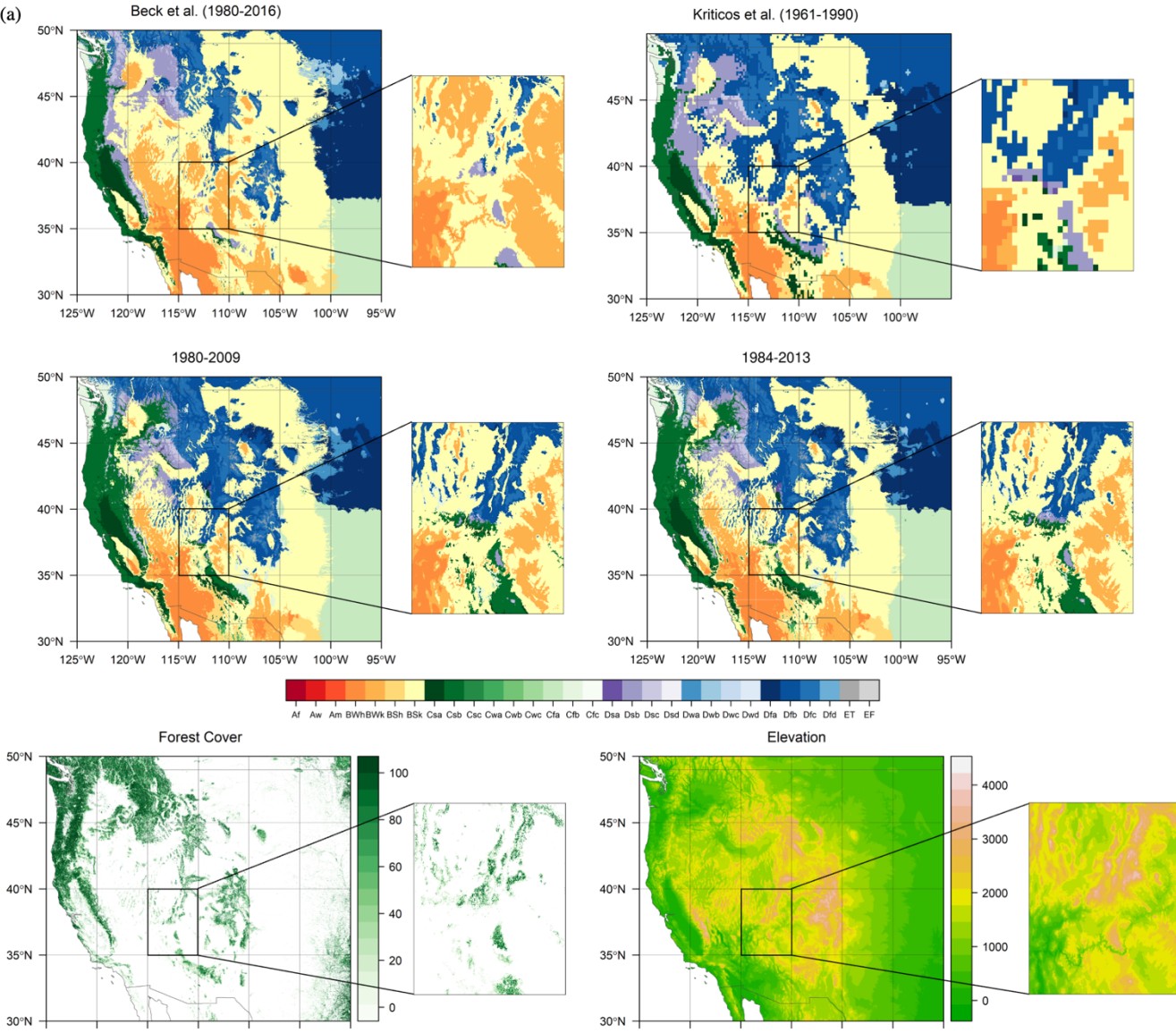

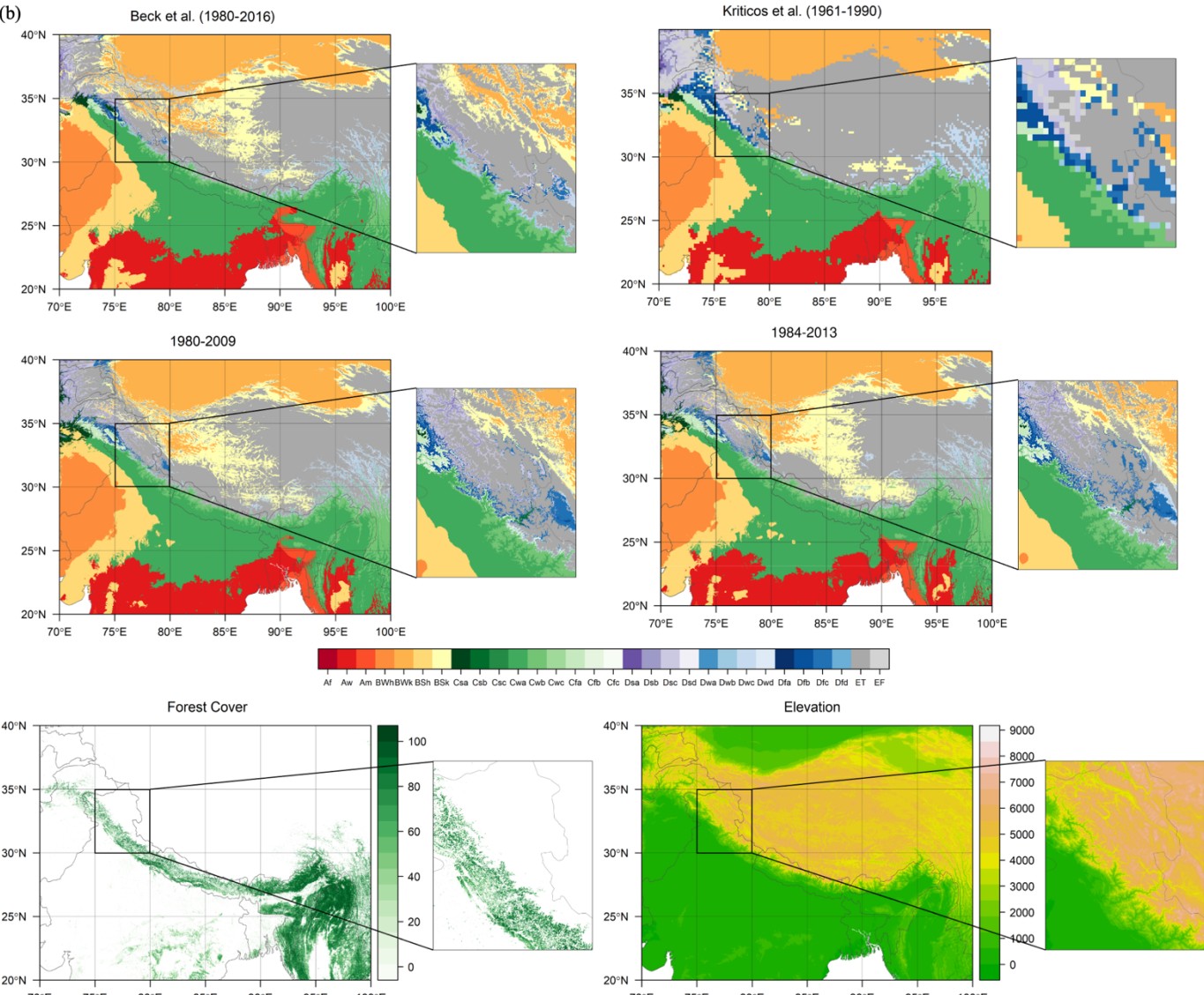

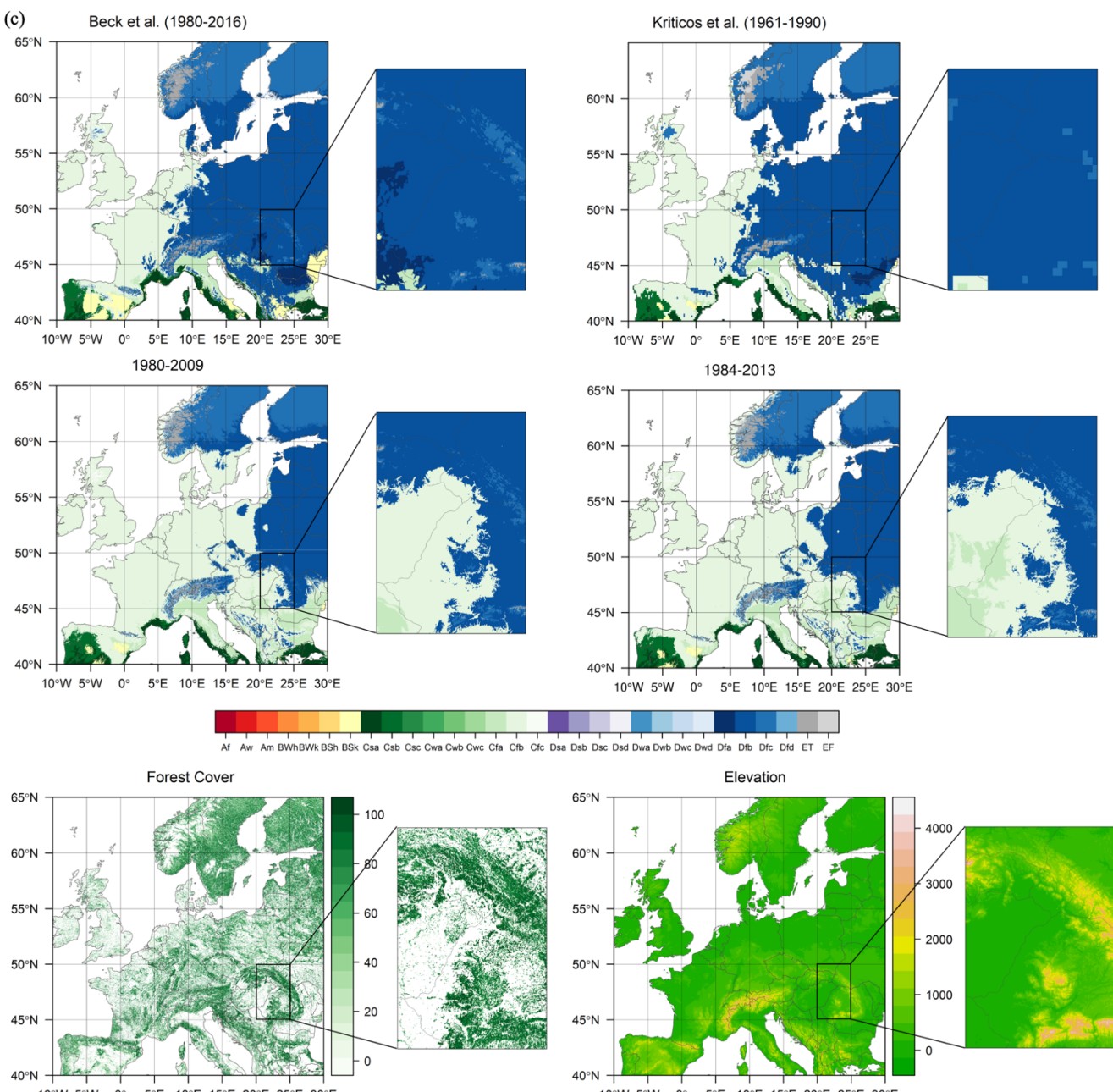

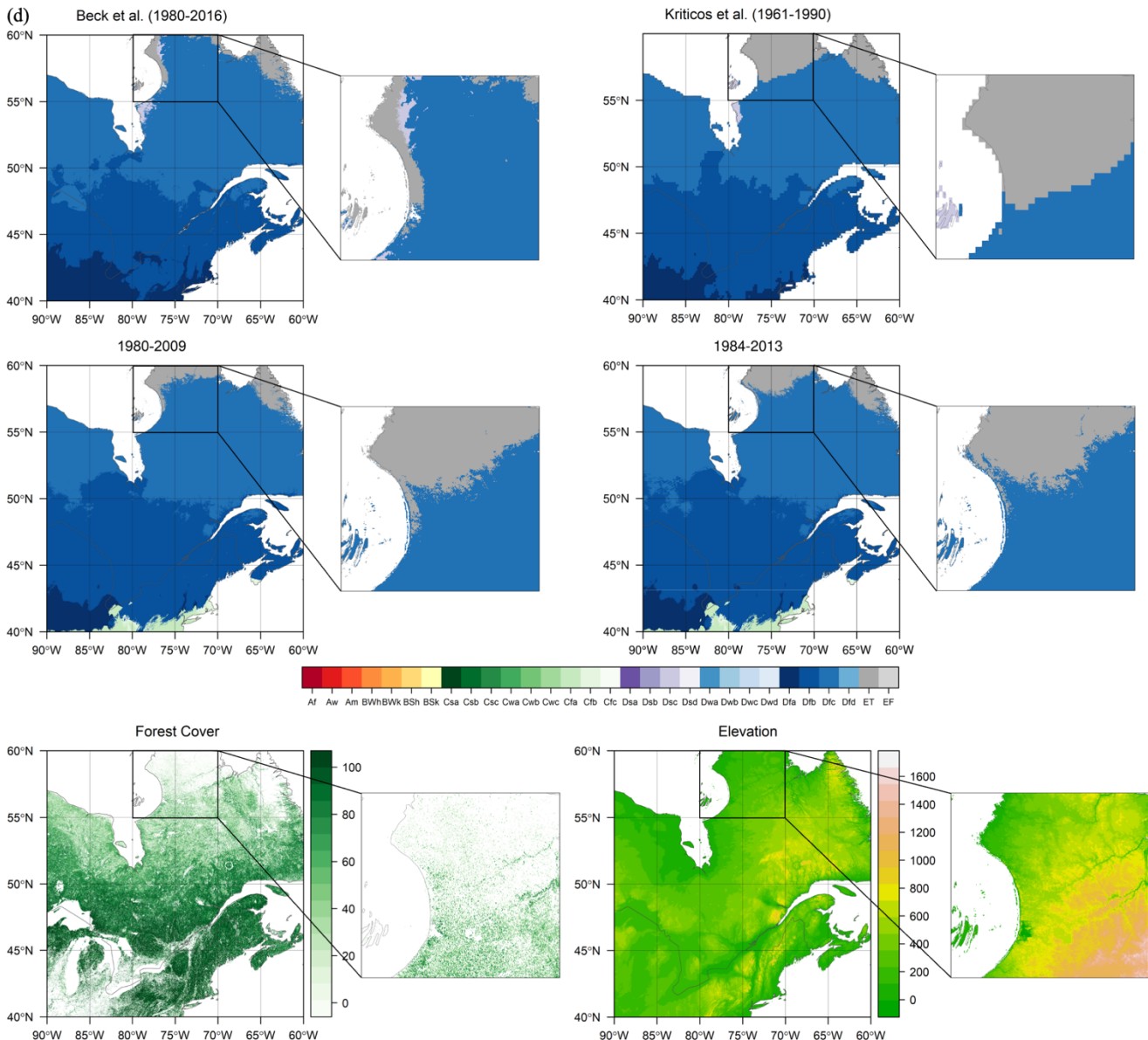

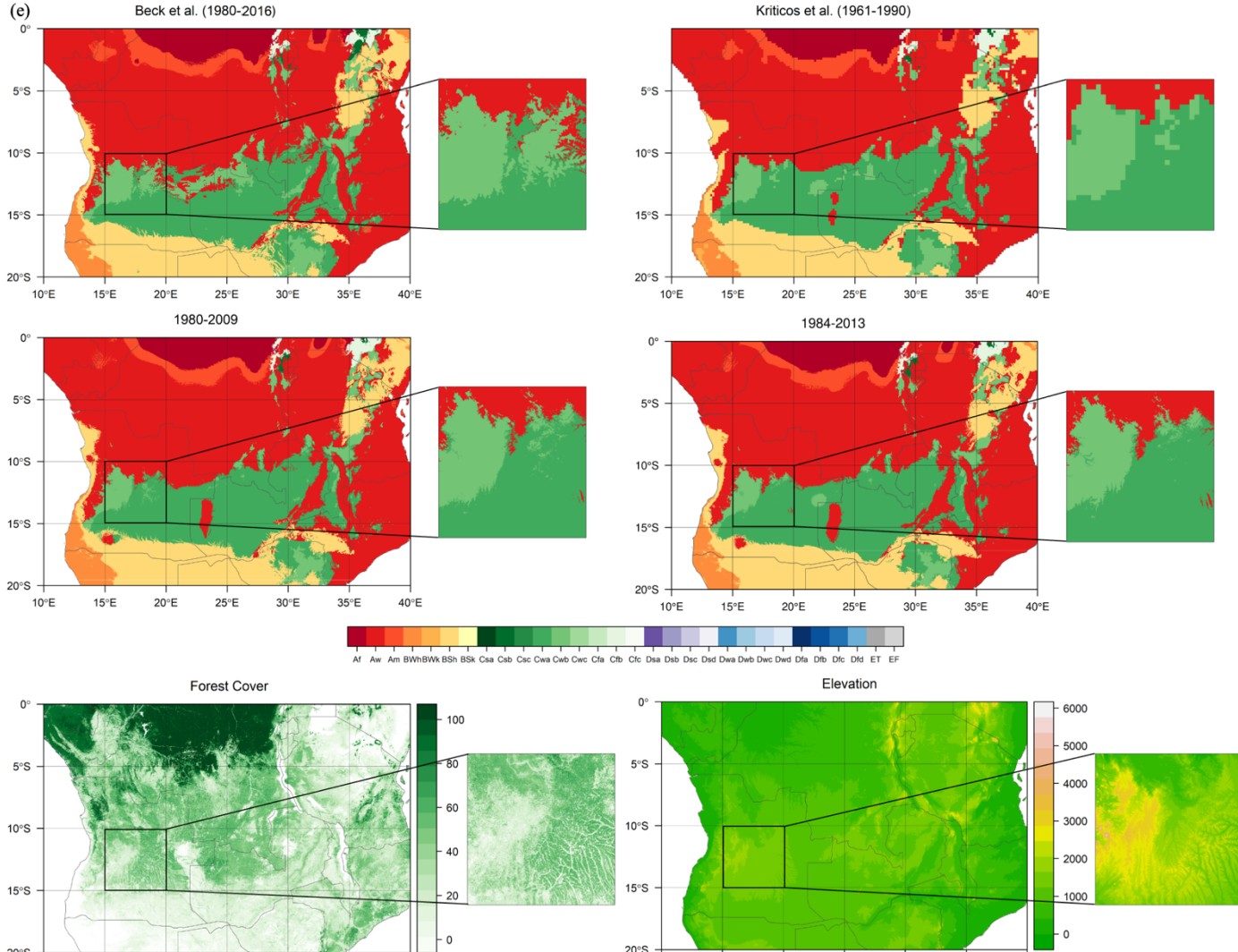

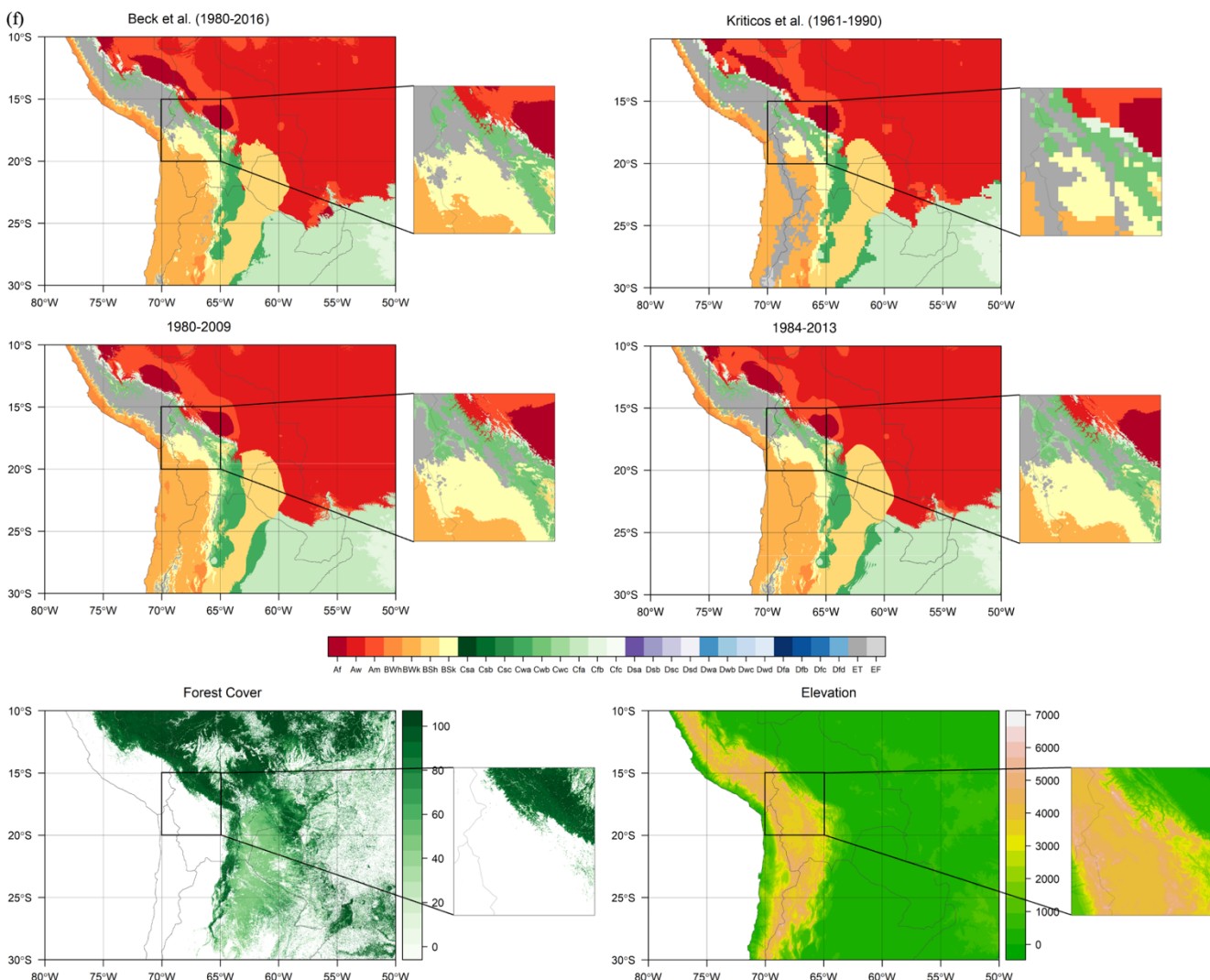

**Figure 8. Köppen-Geiger climate classification maps from previous studies, Beck et al., 2018 (1-km, 1980-2016), and Kriticos et al., 2012 (0.167º, 1961-1990), our study (1-km, 1979-2009 to 1984-2013), associated forest cover and elevation maps, for regions with large spatial gradients in climates or sharp elevation gradients**. (a) central Rocky Mountains, (b) Tibetan Plateau, (c) Europe, (d) high latitudes in North America, (e) Central and eastern Africa, and (f) central Andes. The forest cover map is the 30m Landsat-based forest cover map for year 2000 (Hansen et al., 2013). The elevation data is the NASA SRTM Digital Elevation 30m data (Farr et al., 2007). The representative period of each map is listed in parentheses.

At the regional and continental scale, we compared our Köppen-Geiger climate classification maps with previous map products for regions with large spatial gradients in climates, including central and eastern Africa, Europe, North America, and regions with sharp elevation gradients, including Tibetan Plateau, central Rocky Mountains, central Andes (Fig. 8). We compared the new 1-km Köppen-Geiger climate classification maps from our study for time periods of 1980-2009, and 1984-2013 with the high-resolution Köppen-Geiger maps from two previous studies, Beck et al., (2018), which has a resolution of 1-km and

temporal coverage of 1980-2016, and Kriticos et al., (2012), which has a resolution of 0.0167o and covers 1961-1990. The
Köppen classifications demonstrate good correlation with natural landscape distributions (Belda et al., 2014; Köppen, 1936;
Trewartha, 1954). To show the agreement between the improved Köppen-Geiger climate classification maps and regional
lanscape distributions, we also showed maps of forest cover, and elevation distribution for these regions. Figure 8 illustrate the
enhanced regional details of the maps.

Compared with the Köppen-Geiger climate maps from previous studies with only one time period, the series of the Köppen-
Geiger climate maps from our study demonstrate the ability to capture recent changes in spatial distributions of climate zones.
For example, our maps can detect the significant changes in the climate zones specifically driven by the accelerated global
warming since the 1980s, for example, the poleward movements of boreal (D) and polar (E) climates in high latitudes in North
America shown in the comparison between the 1980-2009 and 1984-2013 Köppen-Geiger climate maps (Fig. 8d). Another
example is the expansion of savanna (Aw) climate into temperature (Cw) climate zone, witnessed in Central Africa (Fig. 8e).

Another improvement of the new series of the Köppen-Geiger climate maps is the application of threshold of -3 oC as the
boundary of temperate (C) and boreal (D) climate zones, which show better agreement with global boreal forest distributions
at regional scale compared with Russell's modification of 0 oC (1931), which Beck et al., (2018), and Kriticos et al., (2012)
utilized. Based on the comparison results of the Köppen climate zones and the biome classifications from the World Wildlife
Federation (Rohli et al., 2015), the boreal (D) climate zone largely corresponds to the distribution of boreal forest (Cui, Liang,
& Wang, 2021). For example, evidenced in Figure 8c, the new Köppen-Geiger climate classification maps from our study
show better agreement with the boreal forest in Carpathian Mountains across Central and Eastern Europe than Beck et al.,
(2018), and Kriticos et al., (2012). Figure 8d also shows good agreement of the northern boundary of boreal (D) climate zone
in northern part of Quebec in Canada with the boundary of Canada's boreal forest.

Moreover, the new Köppen-Geiger maps can show accurate depiction of important topographic features over the regions with
complex topography. For example, the topo-climate of the Himalays southern front determined by the mountain ranges are
represented with more details in the new Köppen-Geiger maps compared with Beck et al., (2018), and Kriticos et al., (2012)
(Fig. 8b). The abrupt changes in climate along the edges of the Andes mountains are also well described in the new maps (Fig.
8f).

In addition, the distribution of tropical (A), temperate (C) and boreal(D) climate zones in the new Köppen-Geiger maps
correspond closely with tree lines in the forest cover maps. The temperate (C) and boreal(D) climate distributions based on the
Köppen-Geiger maps show a better agreemenet with the forest distributions of the Middle and Southern Rocky Mountains
than Beck et al., (2018), and Kriticos et al., (2012) (Fig. 8a). For another example, the boundaries of the tropocal rainforest in
Central Africa and South America are clearly delineated in the in the new Köppen-Geiger maps (Fig. 8e and 8f).

## 4.4 Bioclimatic variables

**Table 5 List of bioclimatic variables derived from downscaled monthly climate data.**

| Bioclimatic Variables | Description |
| --- | --- |
| BIO1 | Annual mean temperature (ºC) |
| BIO2 | Temperature of the warmest month (ºC) |
| BIO3 | Temperature of the coldest month (ºC) |
| BIO4 | Annual precipitation (mm) |
| BIO5 | Precipitation of the warmest half year (mm) |
| BIO6 | Precipitation of the coldest half year (mm) |
| BIO7 | Precipitation of the driest month (mm) |
| BIO8 | Precipitation of the driest month in the warmest half year (mm) |
| BIO9 | Precipitation of the driest month in the coldest half year (mm) |
| BIO10 | Precipitation of the wettest month (mm) |
| BIO11 | Precipitation of the wettest month in the warmest half year (mm) |
| BIO12 | Precipitation of the wettest month in the coldest half year (mm) |

Beyond the Köppen-Geiger climate classification maps, we calculated a set of bioclimatic variables from the monthly climate data (see full list in Table 5). The bioclimatic variables at 1-km sptial resolution can capture regional environmental variations expecially in mountainous areas and areas with strong climate variations. These bioclimatic variables can be used in studies of environmental, agricultural and biological sciences, for example, development of species distribution modeling and assessment of biological impacts induced by climate change. The variables provide descriptions of annual averages, and

seasonality of climates. The warmest half year or the coldest half year is defined as the period of the warmest six months or the coldest six months.

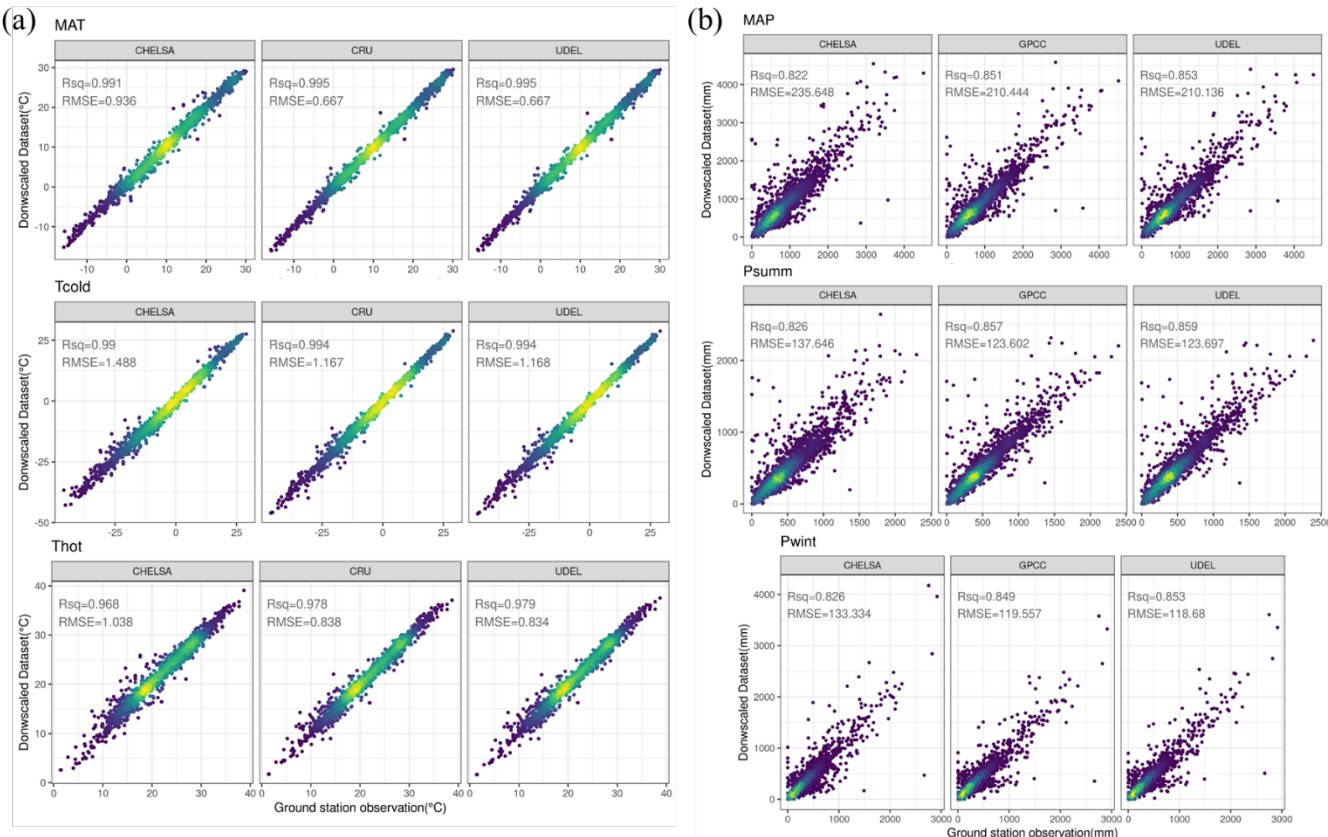

**Figure 9. Scatter plots of the station observations and estimates of bioclimatic variables from downscaled climatology data.** The bioclimatic variables include the 30-yr means of annual temperature (MAT), the air temperature of the coldest month (Tcold), the air temperature of the warmest month (Thot), total annual precipitation (MAP), precipitation of the summer half year (Psumm), and precipitation of the winter half year (Pwint). (a) Scatter plots of the station observations and downscaled temperature data from CHELSA, CRU, UDEL datasets, and (b) and downscaled precipitation data from CHELSA, GPCC, UDEL datasets.

We validated the bioclimatic variables from different datasets with station data from GHCN-D (Menne et al., 2012) and GSOD database (National Climatic Data Center et al., 2015) (Fig. 9). We calculated a linear regression model for the 12 bioclimatic variables for each 10° grid cell (Fig. 10). The 30-yr average mean annual temperature (MAT) from CHELSA dataset shows overall highest fit with station data, with CRU, and UDEL datasets showing smaller, but still strong correlation with station data. The 30-yr average mean annual precipitation (MAP) estimates from GPCC, UDEL, and CHELSA datasets have considerable uncertainties, indicated by relatively low correlation with station observations. In current precipitation datasets, there exist a varied degree of discrepancy in annual estimates over multiple time scales (Sun et al., 2018).

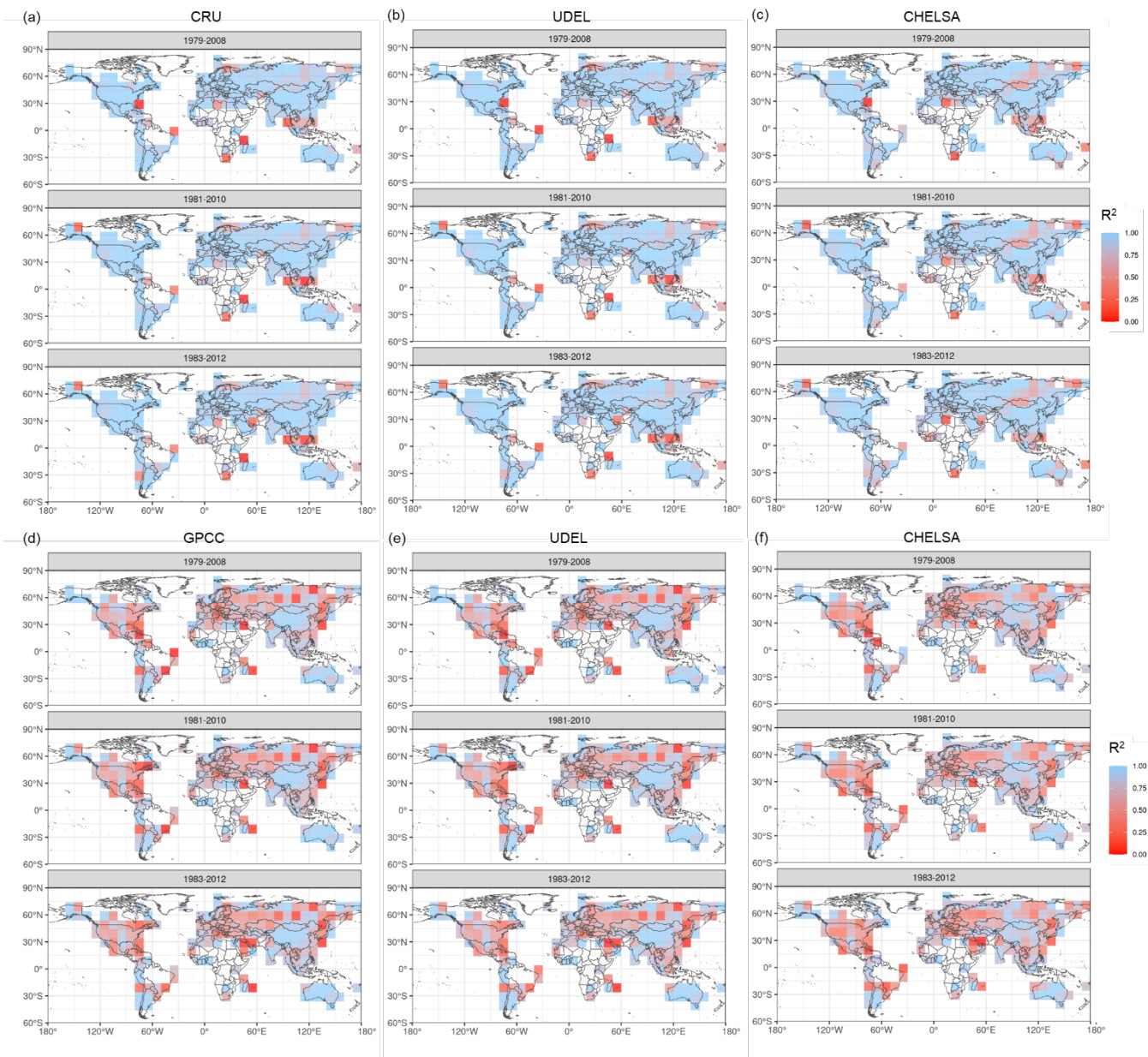

**Figure 10. Small-scale comparison of annual temperature (MAT) and mean annual precipitation (MAP) variables derived from different datasets with station data.** Small-scale correlation between the 30-yr average mean annual temperature (MAT) and mean annual precipitation (MAP) data and ground observations for three historical periods (1979-2008, 1981-2010, 1983-2012). The station data is from GHCN-D and GSOD database. The figure shows the $R^2$ value for 10° grid cells. (a), (b), and (c) are MAT results. (d), (e), and (f) are MAP results. (a) MAT is calculated from downscaled monthly temperature data from CRU dataset, (b) from UDEL dataset and (c) from CHELSA dataset. (d) MAP is calculated from downscaled monthly precipitation data from GPCC dataset, (e) from UDEL dataset and (f) from CHELSA dataset.

## 4.5 Future Köppen-Geiger climate maps

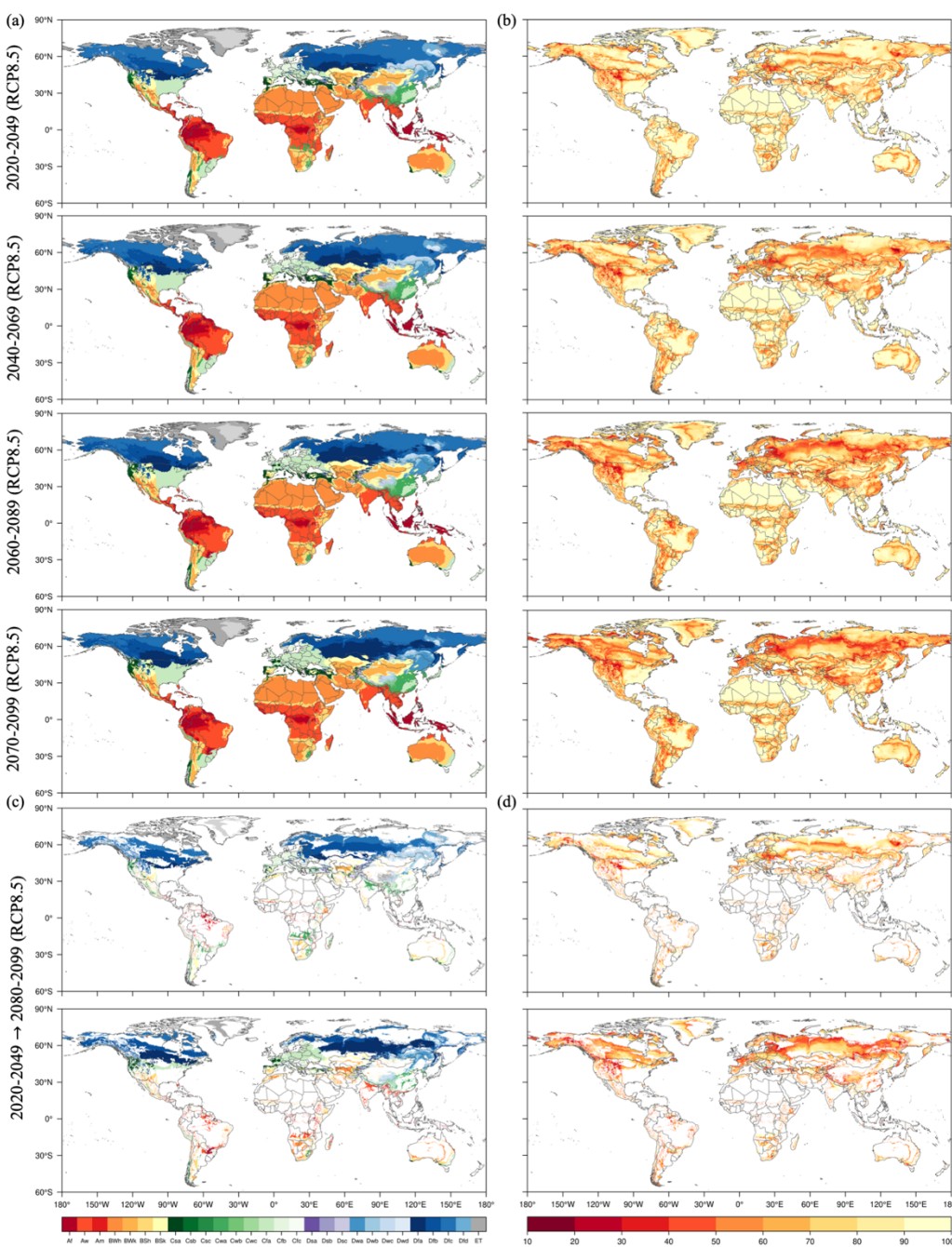

**Figure 11 Global maps of the Köppen-Geiger climate classification for the future periods (2020-2049, 2040-2069, 2060-2089, 2070-2099) under RCP8.5 and associated classification confidence levels.** (a) Future maps of the Köppen-Geiger climate classification and (b) confidence levels associated with the Köppen-Geiger climate classification.

Future Köppen-Geiger climate classification maps under RCP8.5 and associated confidence levels are shown in Figure 11. Indicated by confidence levels, there exist larger uncertainties in the final future climate maps than historical maps, particularly at mid and high latitudes. Climate map for the future period of 2070-2099 shows the largest uncertainty compared with the other future periods.

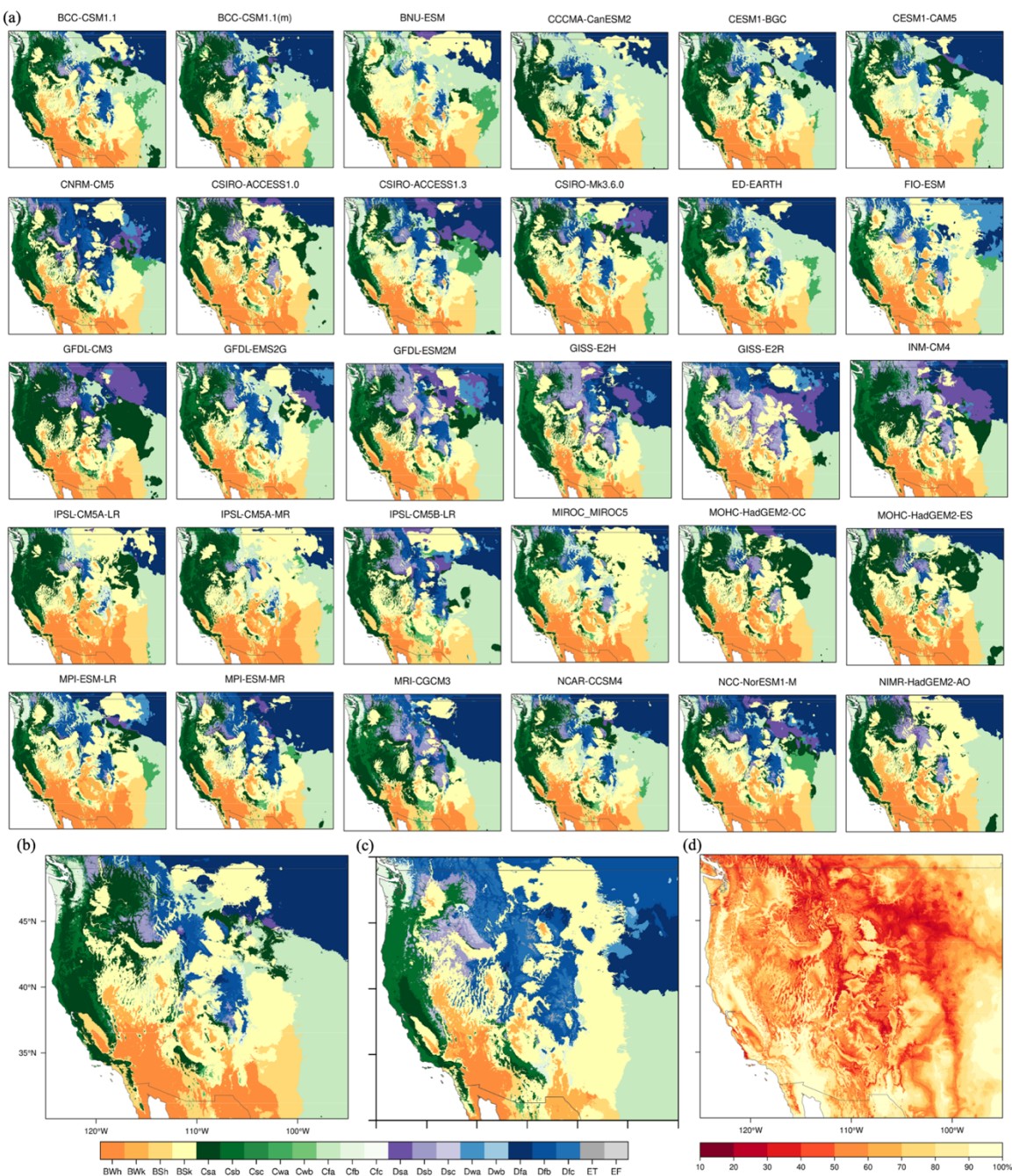

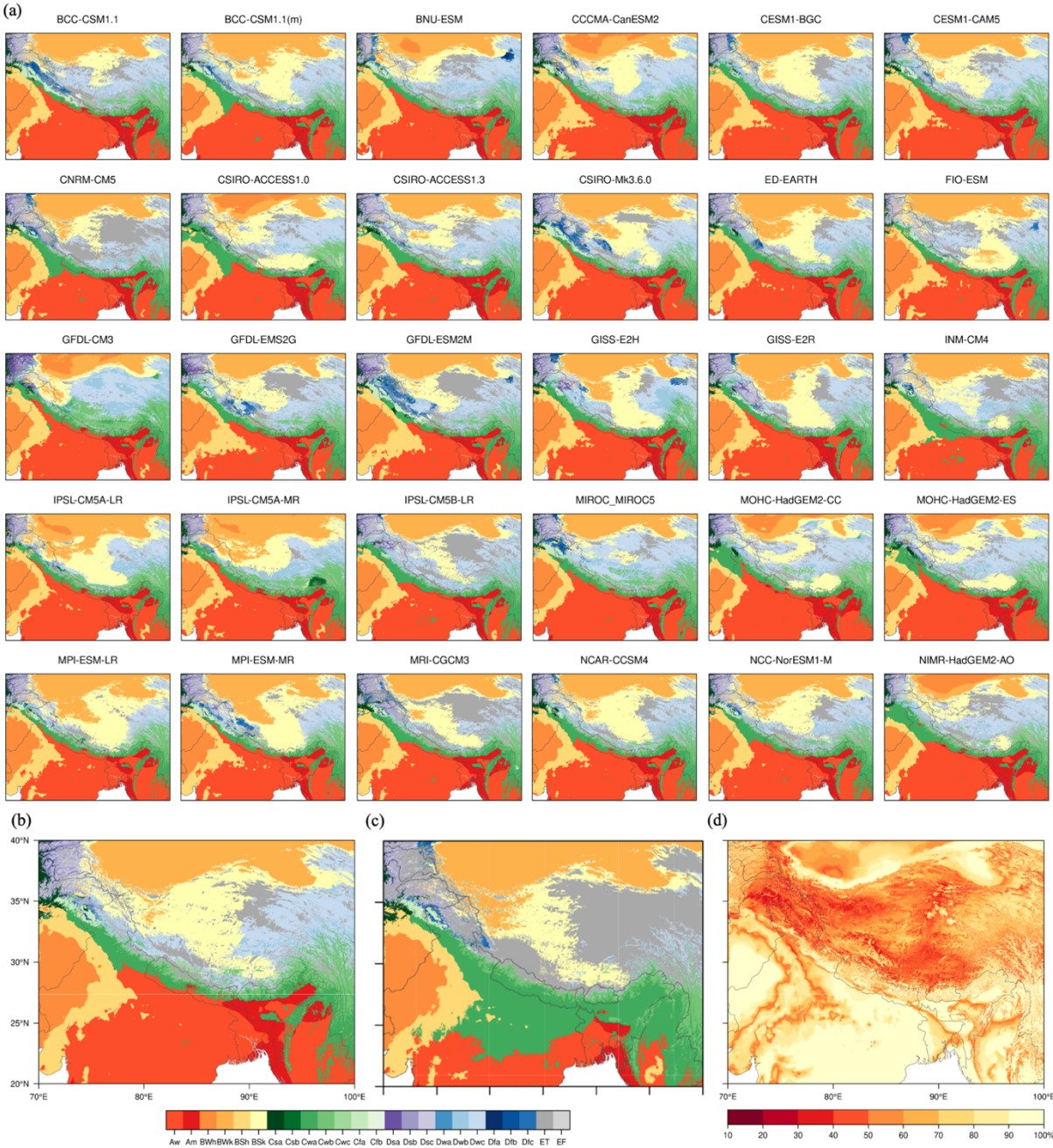

**Figure 13. Future Köppen-Geiger classification and confidence map for 2060-2089 under RCP8.5 with resolution of 1km for the Tibetan Plateau.** (a) Climate maps based on 30 GCMs, (b) the final climate map derived from the most common climate class among all the 30 climate maps, (c) present climate map of 1979-2008, and (d) confidence level distribution of the final climate map.

Future climate classifications derived from the diverse GCM projections for four RCPs, which are inherently uncertain (Gleckler, Taylor, & Doutriaux, 2008; Winsberg, 2012), provide a proxy of global distributions of climatic conditions and can represent potential spatial changes in climate zones under global warming. The large uncertainty and strong disagreement in projected climate classification maps at high latitudes and in regions with rugged terrain can be indicated by relatively low confidence levels. Figure 12 and 13 show the future Köppen-Geiger climate classification maps based on GCM projections under RCP8.5 and associated confidence levels for the central Rocky Mountains and Tibetan Plateau. We generated a future Köppen-Geiger climate classification map for each bias-corrected and downscaled CMIP5 GCM projection (see Fig. 12a and 13a for 2070-2099 maps for the central Rocky Mountains and Tibetan Plateau). Noticeable regional changes in climate zones have been projected by comparing the 2070-2099 and 1979-2008 climate classification maps (see Fig. 12b and 12c for the central Rocky Mountains, and Fig. 13b and 13c for Tibetan Plateau).

### 4.6 Application example: detection of area changes in climate zones

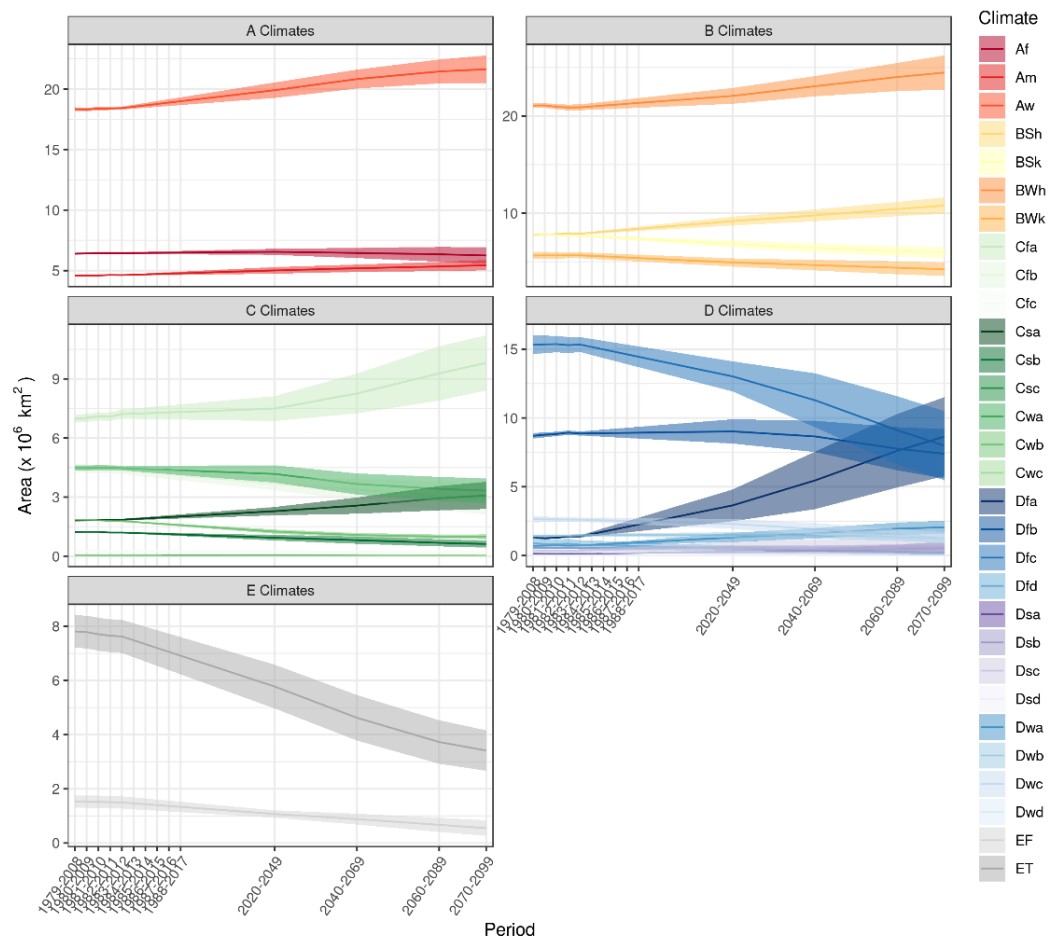

**Figure 14. Area changes in climate zones since the 1980s on a global scale under RCP8.5.** The error bars for historical periods (1979-
2017) indicate standard error in the Köppen-Geiger classification results based on the 9 combinations of observational air temperature and
precipitation datasets and for future periods (2020-2099), the error bars indicate standard error in the Köppen-Geiger classification results
based on the 30 GCMs.

Changes in climatic conditions under global warming have significant impacts on biodiversity and ecological systems. Area
changes of climate zones can indicate spatial shrinkage or expansion of analogous climatic conditions, potentially implying
threats for species range contraction or opportunities for range expansion (Cui, Liang, & Wang, 2021). To examine the area
changes of climate zones, we calculated the total area covered by each climate type for each historical and future periods
under high-emission RCP8.5 scenario (Fig. 14). Our results of changes in area occupied by different climate zones
demonstrate good agreement with results from previous studies (Chan & Wu, 2015). Results show that accelerated
anthropogenic global warming since the 1980s has caused large-scale changes in climate zones and the shifts into warmer
and drier climates are projected in this century. The tropical and arid climates are expanding into large areas in mid latitudes
whereas the high-latitude climates will experience significant area shrinkage.

## 5 Conclusion

Changes in broad-scale climatic conditions, driven by anthropogenic global warming, lead to the redistribution of species
diversity and the reorganization of ecosystems. Distributions of the Earth's climatic conditions have been widely characterized
based on the Köppen climate classification system. The Köppen climate classification maps require fine resolutions of at least
1-km to detect relevant microrefugia and promote effective conservation. Studies examining recent and future interannual or
interdecadal changes in climate zones at regional scale needs more accurate depiction of fine-grained climatic conditions,
continuous and longer temporal coverage.

We presented an improved long-term Köppen-Geiger climate classification map series for six historical 30-yr periods in 1979-
2013 and four future 30-yr periods in 2020-2099 under RCP2.6, 4.5, 6.0 and 8.5. To improve the classification accuracy and
achieve a resolution as fine as 1-km, we combined multiple datasets, including WorldClim V2, CHELSA V1.2, CRU TS v4.03,
UDEL, GPCC datasets and bias-corrected downscaled CMIP5 model simulations from CCAFS. The historical climate maps
are based on the most common climate type from an ensemble of climate maps derived from combinations of observational
climatology datasets. The future climate maps are based on an ensemble of climate maps derived from 35 GCMs. We estimated
the corresponding confidence levels to quantify the uncertainty in climate maps. We also calculated 12 bioclimatic variables
at the same 1-km resolution using these climate datasets for the same historical and future periods to provide data of annual
averages, seasonality, and stressful conditions of climates.

To validate the Köppen-Geiger climate classification maps, we used the station observations from GHCN-D and GSOD
database. Our validation results show that the new Köppen-Geiger maps have comparatively higher overall accuracy than all
the previous studies. Although the new maps exhibit improved overall accuracy, relatively lower confidence level and larger

discrepancy in classification results are found especially in mountainous regions and major climate transitional zones located in mid and high latitudes. The confidence levels can provide a useful quantification of classification uncertainty.

Compared with climate maps from previous studies with a single present-day period, the series of the Köppen-Geiger climate maps from our study demonstrate the ability to capture recent and future projected changes in spatial distributions of climate zones. On regional and continental scale, the new maps show accurate depictions of topographic features and correspond closely with vegetation distributions. Our Köppen-Geiger climate classificaion maps can offer a descriptive and ecological relevant way to provide insights into changes in spatial distributions of climate zones.

One of the limitations is that the future Köppen-Geiger climate maps built on dowscaled climate model projections exsit unaviodable uncertainties. The classification agreement levels of GCMs are relatively low at high latitudes and in regions with rugged terrain. The main sources of model discrepancies and uncertainties are deficiencies in model physics and varied model resolution. The climate model outputs have coarse spatial resolution varying from 70-400 km and cannot well represent future climate change at the same scale of 1-km as our baseline climatology. Through bias-corretion and donwscaling methods, we made assumptions that local relationships between climatic variables remain constant across different scales, leading to a compromise between spatial scale and climate model physics.

We also tested the sensitivity of classification results to different time scale, dataset input, and data integration methods. Results show that 30-yr time scale exhibited the highest accuacy results. Moreover, using the mean of multiple datasets from CHELSA, CRU, UDEL, and GPCC could lead to better classification results. Last, we provided a heuristic example which used climate classification map series to detect the long-term area changes of climate zones, showing how the new Köppen-Geiger climate classification map series can be applied in climate change studies. With improved accuracy, high spatial resolution, long-term continuous time coverage, this global dataset of the Köppen-Geiger climate classification and bioclimatic variables can be used to in conjunction with species distribution models to promote biodiversity conservation, and to analyse and identify recent and future interannual or interdecadal changes in climate zones on a global or regional scale.

**Data Availability**

This high-resolution global dataset of the Köppen-Geiger climate classification and bioclimatic variables dataset for historical periods in 1979-2013 is available at http://doi.org/10.5281/zenodo.5347837 (Cui, Liang, Wang, & Liu, 2021b). The dataset for future periods in 2020-2100 is available at http://doi.org/10.5281/zenodo.4542076 (Cui, Liang, Wang, & Liu, 2021a).

**Author Contribution**

C.D. designed the computational framework, performed data collection and processing, conducted validation and sensitivity analyzes, and wrote the manuscript. L.Z. contributed to the data processing. L.S. was involved in planning and supervised the work. All authors discussed the results and commented on the manuscript.

**Competing Interests**

The authors declare that they have no conflict of interest.

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
