# Peer review of "A 1-km global dataset of historical (1979-2013) and future (2020-2100) Köppen-Geiger climate classification and bioclimatic variables"

_Earth System Science Data, 2021_

## Author Comment (AC1)

**Responses to Referee#1**

We thank Referee#1 for the careful review of the manuscript and the constructive comments. We have revised the manuscript based on the comments and provided detailed point-by-point responses to the comments. Our replies are highlighted in blue.

Cui et al. present Köppen-Geiger climate classification maps for ten 30-yr historical periods between 1979-2017 and future periods under different RCP scenarios. The historical maps were derived from combinations of 3 global temperature products and 3 global precipitation products, while the future maps were derived from bias-corrected CMIP5 projections. The authors claimed that their maps can capture recent and future changes in spatial distribution of climate zones. This dataset is useful and relevant for wide audience, but there is an important issue that have to be addressed before publication:

The authors compare two maps for 1980-2009 and 1987-2016 in Fig. 11. However, according to Table 1 and Table 3, the map for 1980-2009 was based on 3 temperature products (CRU, UDEL and CHELSA) and 3 precipitation products (UDEL, CHELSA, GPCC), while the map for 1987-2016 was based on 2 temperature products (CRU and UDEL) and 2 precipitation products (UDEL and GPCC), as CHELSA only covers 1979-2013 (Table 1). Evidenced by Fig. 5, the CHELSA precipitation have a large impact on the KGC map. It is also be seen in Fig. 14, where the abrupt changes are all happening between 1983-2012 and 1984-2013. So I suspect the difference for the two periods largely come from inconsistent inputs for the two periods, and is not a reflection of the true shift in climate zones. I strongly recommend the authors to discuss the impact of having CHELSA before 2013 and not having CHELSA afterwards on their time series of KGC map.

Thank you for pointing out the issue caused by the inconsistent data inputs for the historical periods. To address it, we excluded the KGC maps based on the 2 temperature products (CRU and UDEL) and 2 precipitation products (UDEL and GPCC) and kept the KGC maps based on 3 temperature products (CRU, UDEL and CHELSA) and 3 precipitation products (UDEL, CHELSA, GPCC) as our final climate map product. According to the evaluation results, the final map product based on CHELSA, CRU, GPCC and UDEL data exhibits higher overall classification accuracy than the maps based on CRU, GPCC and UDEL data. Additionally, the CHELSEA precipitation data considered topo-climatic drivers and predicted better precipitation patterns (Karger et al., 2017), which may not be well captured in statistically downscaled 0.5° climatology data. Having CHELSA as data as input of climate maps can slightly improve the overall classification accuracy and present more accurate depiction of spatial patterns. Therefore, we decided to use the KGC maps based on CHELSA, CRU, UDEL and GPCC as our final climate map product. But since the CHELSA climatology data has a temporal span from 1979-2013, the final KGC map product will cover six historical periods, including 1979-2008, 1980-2009, 1981-2010, 1982-2011, 1983-2013, and 1984-2013. We have updated it in the revised manuscript and created a new version of the historical KGClim dataset.

There are also some minor issues:

1. A grammar mistake in L17: "The new maps offer higher classification accuracy", higher than which products?

Thank you for this comment. Based on out validation results, our new historical climate maps demonstrate higher overall classification accuracy than all the other existing climate map

products. We have added the detail and revised the sentence to "The new maps offer higher classification accuracy than existing climate map products".

2. L30 and L36: the authors repeated the definition of the Koppen classification in these two lines, but used "annual cycles" in L30 and "seasonal cycles" in L36. What are the difference?

Thank you for pointing out the issue of the use of "annual cycles" and "seasonal cycles" regarding the definition of the Köppen classification. The Köppen classification is defined based on the seasonal phase of temperature and precipitation annual cycles. Using "annual cycles" and "seasonal cycles", we refer to the seasonality of temperature and precipitation. To clarify the difference, we have changed "annual cycles" to "seasonal phase of annual cycles".

3. L103-L104: "Evaluation results indicated that incorporating only CRU, UDEL temperature datasets and UDEL, GPCC precipitation datasets led to higher accuracy in the classification results." Table 3 tells me the combinations of CRU, UDEL and CHELSA temperature and UDEL, GPCC and CHELSA precipitation lead to the highest accuracy. So I don't understand why this sentence here did not mention CHELSA.

Thanks for your comment. To decide which datasets to use to generate the historical KGC maps, we first tested several observational climatology datasets with coarse resolution of 0.5°, including CRU, UDEL, GHCN\_CAMS, GPCC, UDEL, and PREC/L. We found out that using only CRU, UDEL temperature datasets, and the UDEL, GPCC precipitation datasets have better classification results than having all the CRU, UDEL, GHCN\_CAMS, GPCC, UDEL, GHCN\_CAMS and PREC/L datasets as input. To clarify this, we have revised the sentence, "Evaluation results indicated that incorporating only CRU, UDEL temperature datasets and UDEL, GPCC precipitation datasets and excluding GHCN\_CAMS and PREC/L datasets led to higher accuracy in the classification results". The paragraph following the sentence introduced the CHELSA dataset and explained the reasons for using the 1-km CHELSA and WordClim datasets in addition to the 0.5° datasets to correct topographic effects and provide better description of precipitation patterns. Table 3 shows the accuracy results with or without CHELSA dataset as input or using CHELSA data alone.

4. L120: Please give the ref.

Thanks for this comment. We have provided the reference, "Technical evaluation showed that the bias-correction method that CCAFS data applied reduced climate model bias by 50–70%, which could potentially address the bias issue in model simulations for the threshold-based Köppen classification scheme (Navarro-Racines, Tarapues, Thornton, Jarvis, & Ramirez-Villegas, 2020)."

 L123: The Köppen climate classification scheme was first introduced in 1884 (Rubel, F. & Kottek, M. Comments on: The thermal zones of the Earth by Wladimir Köppen (1884). (2011) doi:10.1127/0941-2948/2011/0258.)

We have corrected the year to 1884 and added the reference in the revised manuscript.

Köppen, W. (1884). Die Wärmezonen der Erde, nach der Dauer der heissen, gemässigten und kalten Zeit und nach der Wirkung der Wärme auf die organische Welt betrachtet. Meteorologische Zeitschrift, 1(21), 5–226.

6. L125: KGC is not explained for the first time being used.

Thanks for pointing it out. We have added the explanation of KGC for its first time mentioned in the manuscript.

7. 10. The accuracy of Beck et al. (2018) not plotted.

We have added the accuracy of Beck et al. (2018) in Figure 10.

8. L288: "Duplicate stations in the two datasets were further removed." This sentence should be moved to section 4.3?

We accepted the suggestion and moved the sentence to section 4.3.

9. L306-307: If the products here are better than previous one, how the previous "worse" maps can be used for "evaluation"?

Thanks for raising the concern regarding the evaluation of the map product. We used the previous climate maps in addition to forest cover and elevation maps for regional and continental scale comparison to identify the potential improvements of the map product. The expression of "evaluation" is not appropriate and we have changed the sentence to "We compared the new Köppen-Geiger climate classification maps with the high-resolution Köppen-Geiger maps from two previous studies, Beck et al., (2018), and Kriticos et al., (2012)."

10. L315-317: "Another improvement ..., which show better agreement with global boreal forest distributions". I do not find the evidence from the figure.

Thank you for your comment. We updated Figure 11 to show the continental and regional scale comparison with more details of the Köppen-Geiger climate classification maps from previous studies, Beck et al., 2018 (1-km, 1980-2016), and Kriticos et al., 2012 (0.167°, 1961-1990), our study (1-km, 1979-2009 to 1984-2013), forest cover and elevation maps. Shown in the updated figure below for Europe, we can see that the new Köppen-Geiger climate classification maps from our study show better agreement with the boreal forest in Carpathian Mountains across Central and Eastern Europe at small scale. The reason causing the different boreal (D) climate zone distribution compared with Beck et al., (2018), and Kriticos et al., (2012), is that we followed the Köppen-Geiger climate classification as described in Kottek, Grieser, Beck, Rudolf, and Rubel (2006), and Rubel and Kottek (2010), and used the threshold of -3 °C as the boundary of temperate (C) and boreal (D) climate zones while Beck et al., (2018), and Kriticos et al., (2012) used the Russell's modification (1931), which is based the distributions of topographical features and vegetation in western United States. We have added the explanation and description of the figure in the revised manuscript.